# The Opto-inflammasome in zebrafish as a tool to study cell and tissue responses to speck formation and cell death

Eva Hasel de Carvalho[1], Shivani S Dharmadhikari[1], Kateryna Shkarina[2], Jingwei Rachel Xiong[3], Bruno Reversade[4,5,6,7], Petr Broz[2], Maria Leptin[1]*

[1]European Molecular Biology Laboratory, Heidelberg, Germany; [2]Department of Biochemistry, University of Lausanne, Lausanne, Switzerland; [3]Skin Research Laboratories, A*STAR, Singapore, Singapore; [4]Institute of Molecular and Cell Biology, A*STAR, Singapore, Singapore; [5]Department of Pediatrics, National University of Singapore (NUS), Singapore, Singapore; [6]Medical Genetics Department, Koç University School of Medicine (KUSOM), Istanbul, Turkey; [7]Laboratory of Human Genetics & Therapeutics, Genome Institute of Singapore (GIS), A*STAR, Singapore, Singapore

**Abstract** The inflammasome is a conserved structure for the intracellular detection of danger or pathogen signals. As a large intracellular multiprotein signaling platform, it activates downstream effectors that initiate a rapid necrotic programmed cell death (PCD) termed pyroptosis and activation and secretion of pro-inflammatory cytokines to warn and activate surrounding cells. However, inflammasome activation is difficult to control experimentally on a single-cell level using canonical triggers. We constructed Opto-ASC, a light-responsive form of the inflammasome adaptor protein ASC (Apoptosis-Associated Speck-Like Protein Containing a CARD) which allows tight control of inflammasome formation in vivo. We introduced a cassette of this construct under the control of a heat shock element into zebrafish in which we can now induce ASC inflammasome (speck) formation in individual cells of the skin. We find that cell death resulting from ASC speck formation is morphologically distinct from apoptosis in periderm cells but not in basal cells. ASC-induced PCD can lead to apical or basal extrusion from the periderm. The apical extrusion in periderm cells depends on Caspb and triggers a strong $Ca^{2+}$ signaling response in nearby cells.

*For correspondence:
mleptin@uni-koeln.de

Competing interest: The authors declare that no competing interests exist.

**Editor's evaluation** This important study describes a novel optogenetic approach to study the responses to inflammasome activation and cell death in zebrafish. The validation of the model is compelling, and its application reveals novel insights in the response to cell death. This new model should help further advance future efforts centered on cell death responses in vivo.

## Introduction

The innate immune system plays a crucial role in the early recognition and eradication of potentially dangerous microorganisms. Inflammasomes are among the best characterized components of innate antimicrobial defense in vertebrates. Inflammasome activation is induced following recognition of pathogen-associated molecular patterns (PAMPs) or danger-associated molecular patterns (DAMPs) by intracellular pattern-recognition receptors, such as NOD-like receptors (NLRs) and AIM2-like receptors (ALRs). This leads to the formation of large multiprotein platforms, which often contain

the adaptor molecule ASC (apoptosis-associated speck-like protein containing a CARD), that can recruit and activate proinflammatory caspases (*Martinon et al., 2002*). These caspases cleave and thereby activate downstream effector molecules, including the pore-forming Gasdermins (GSDMs) and pro-inflammatory cytokines such as interleukin-1β (IL-1β) and interleukin-18 (IL-18), which are then released through the GSDM pores (*Evavold et al., 2018*; *Heilig et al., 2018*).

Inflammasome pathway components are expressed in immune and non-immune cells, but have been studied mostly in cultured cells or bone marrow derived macrophages (see e.g. *Tweedell et al., 2020*). Recent studies also highlight their involvement in the antimicrobial defense in epithelial tissues (*Santana et al., 2016*; *Churchill et al., 2022*).

Epithelia are the first point of contact between the host organism and microbial invaders. Although the role of inflammasome components can be studied in vivo using epithelial infection models, little is known about the direct effects of inflammasome formation on cells and their neighbours in the context of the live tissue. Thus, analysis of this dynamic and rapid process would benefit from precise spatial control of inflammasome formation and live imaging of the ensuing physiological events.

Zebrafish are well suited to study innate and adaptive immunity in vivo (*Novoa and Figueras, 2012*; *Renshaw and Trede, 2012*). Their transparency in early stages and a multitude of fluorescent reporter lines offer ideal conditions for real-time visualization of cellular processes in tissues. Some of the genes encoding core components of innate immune signaling pathways are highly conserved in the zebrafish (*Stein et al., 2007*), while others, like those encoding the fish-specific NLR proteins, are more divergent (*Howe et al., 2016*). The adaptor protein ASC is among the most highly conserved inflammasome components in all vertebrates. ASC consists of a pyrin domain (PYD) and a caspase recruitment domain (CARD) connected by a flexible linker. Once activated, ASC undergoes prion-like oligomerization, concentrating the entire pool of ASC in the cell in one spot to form a dense fibrous structure, the ASC speck (*Dick et al., 2016*).

In the zebrafish, ASC is expressed both in epithelial and immune cells, such as macrophages and neutrophils (*Kuri et al., 2017*). The larval zebrafish skin consists of two cell layers. The outer cell layer, the periderm, consists of tightly connected keratinocytes with characteristic actin ridges; the underlying layer consists of basal cells which in adult zebrafish harbor a stem cell pool to replace dying cells in the skin (*Lee et al., 2014*). During larval development the periderm is gradually replaced and until then grows by divisions and asymmetric fission (*Chan et al., 2022*). A functional inflammasome can be experimentally induced in both layers of the larval zebrafish skin by over-expressing ASC-mKate2, leading to immediate cell death (*Kuri et al., 2017*).

As in mammals, caspases in the zebrafish are recruited to the ASC speck (*Kuri et al., 2017*). The zebrafish genome encodes three inflammatory caspases which contain a PYD domain rather than the CARD found in other vertebrates, Caspa, Caspb (*Masumoto et al., 2003*) and Casp19b (*Spead et al., 2018*) to the ASC speck. The functional homologues of mammalian Caspase-1 in zebrafish, Caspa and Caspb, are both able to induce cell death in zebrafish (*Kuri et al., 2017*; *Shkarina et al., 2022*), and GSDMD cleavage and pyroptosis in human cells (*Shkarina et al., 2022*). Caspb has been shown to cleave the zebrafish Gasdermins, Gsdmea and Gsdmeb, in vitro (*Chen et al., 2021*), while Caspa has been shown to co-localize with ASC-specks (*Masumoto et al., 2003*; *Kuri et al., 2017*).

Upon cleavage, GSDMs assemble into pores in the plasma membrane and lead to the release of the cell's cytosol, including inflammatory cytokines (*Kayagaki et al., 2015*; *Shi et al., 2015*; *Liu et al., 2016*; *Sborgi et al., 2016*), a hallmark of pyroptotic cell death (*Rühl and Broz, 2022*). In the absence of GSDMD, mammalian caspase-1 activates apoptosis as a default pathway (*Heilig et al., 2020*; *Tsuchiya et al., 2021*), and the same was found for ASC-containing inflammasomes in the absence of caspase-1 (*Sagulenko et al., 2013*; *Kitazawa et al., 2017*; *Lee et al., 2018*). Apoptosis is morphologically distinct from pyroptosis and characterized by preservation of membrane integrity, characteristic membrane blebbing and progressive fragmentation of the cell, in contrast to pyroptosis, which most commonly leads to a characteristic cell swelling.

We have previously developed optogenetic variants for human and zebrafish caspases which can be activated by light-induced oligomerization of Cry2olig, a photosensitive protein that undergoes rapid homo-oligomerization in response to blue light (*Taslimi et al., 2014*; *Shkarina et al., 2022*). This enabled us to selectively induce multiple forms of programmed cell death (PCD) in different cell types in vitro, in organoids and in live zebrafish larvae. We observed that zebrafish periderm cells are apically

extruded from their surrounding epithelium upon stimulation of the inflammatory caspases but are basally extruded after activation of the apoptotic caspase-8 (*Shkarina et al., 2022*).

To further study inflammasome formation and the resulting cell death in the zebrafish skin upstream of inflammatory caspases, we have now generated an optogenetic variant of zebrafish ASC (Opto-ASC), which efficiently induces speck formation in single cells in both layers of the larval skin. Opto-ASC specks cause inflammatory cell death followed by apical or basal extrusion in periderm cells, which is morphologically distinct from apoptosis. ASC-speck-induced cell death requires Caspa and Caspb but we could not confirm the role of GSDMs in this process; we therefore refrain from calling it pyroptosis. We show that Opto-ASC specks efficiently trigger cell death in the periderm via inflammatory apical or basal extrusion.

## Results

### Optogenetic activation of ASC-oligomerization

In the past, in vivo induction of inflammasome assembly has been achieved only via overexpression of ASC or NLRs (*Kuri et al., 2017*), or by treatment with proinflammatory chemicals like $CuSO_4$ (*Kuri et al., 2017*) or pathogens (*Tyrkalska et al., 2016*; *Li et al., 2018*; *Forn-Cuní et al., 2019*). However, such chemical and bacterial treatments affect the entire organism, which complicates targeted and controlled studies of cellular and tissue responses. In contrast, the over-expression of ASC efficiently triggers speck formation and cell death, but does not allow targeting these processes to predetermined cells. To overcome these limitations and to enable specific induction of inflammasome assembly with spatial and temporal control, we tagged zebrafish ASC with Cry2olig to produce 'Opto-ASC'. Cry2olig is a protein module that, upon exposure to blue light, reversibly oligomerizes within seconds (*Taslimi et al., 2014*). We fused the mCherry-Cry2olig domain to the N-terminal PYD of ASC and placed this cassette under the control of a heat shock responsive element (HSE) (*Bajoghli et al., 2004*), to be able to selectively induce expression in experimental larvae. We generated stable zebrafish lines carrying the plasmid shown in *Figure 1A* using Tol2-based random genome integration. The plasmid contains a screening cassette in which tagRFP is expressed under the heart-specific myosin light chain 2 promoter (cmlc2). Larvae were screened for red fluorescent hearts at 2.5 days post fertilization (dpf).

Opto-ASC larvae were heat-shocked to induce expression and shielded from light to prevent uncontrolled Opto-ASC activation. The photoactivation and imaging experiments were performed at 3dpf as shown in *Figure 1B*.

The expression of Opto-ASC became detectable around 3 hr after the heat shock and reached a maximum at around 12 hr, with mosaic expression of Opto-ASC in single cells of all tissues. Skin cells of different layers can be identified by their distinct morphology. Cells in the periderm, the outer layer of skin cells, have sharp boundaries and a characteristic pattern of actin ridges in the apical cortex. The underlying basal cells are smaller and have less well-defined boundaries (*Figure 1C*). In absence of blue light stimulation, Opto-ASC did not form specks, but did so efficiently when stimulated with a 488 nm confocal laser (*Figure 1D*, *Video 1*).

The exposure of cells expressing Opto-ASC to 488 nm laser light led to the formation of a speck within minutes followed by immediate death of the cell in both epidermal layers (*Figure 1E*, *Video 2*). Cells in the periderm were usually apically extruded from the tissue within minutes after the appearance of the speck. Even when small regions were selectively illuminated with the 488 nm laser, the specks also appeared in regions outside the illuminated area, and sometimes even in the entire larva, likely due to the out-of-plane Cry2olig activation and confocal light diffusion. This could be avoided by using a two-photon laser, which induced Opto-ASC oligomerization in single cells without affecting surrounding cells (*Figure 1F*, *Video 3*).

In muscle cells, illumination induced the formation of multiple stable oligomers of Opto-ASC without causing cell death (*Figure 1—figure supplement 1A*), as also previously shown for ASC-mKate2 specks (*Kuri et al., 2017*). These Opto-ASC specks remained stable and did not disperse after light exposure was terminated. As control for the contribution of the ASC moiety of the construct to oligomer formation we injected a construct with a mutant variant of Cry2olig, OptoR489E-ASC, which formed no oligomers and did not induce cell death (*Figure 1—figure supplement 1B*).

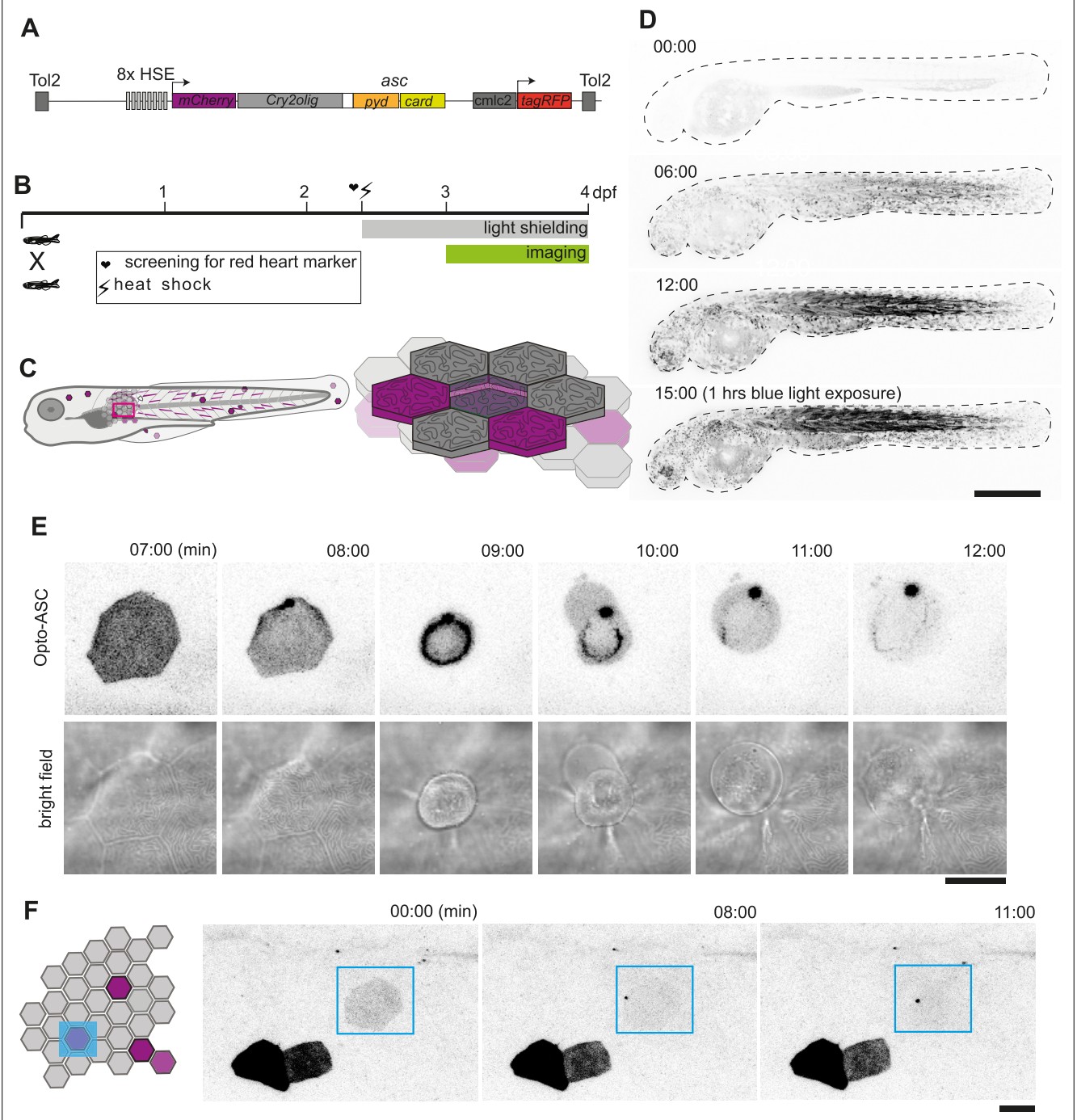

**Figure 1.** Optogenetic tools for ASC-dependent inflammasome formation. (**A**) The optogenetic construct (Opto-ASC) consists of a heat shock element placed upstream of a cassette containing the sequences for *mCherry*, the photoactuator *Cry2olig* domain and *asc,* followed by a cassette containing the "red heart" marker cmlc2:*tagRFP*. The two cassettes are placed between Tol2 sites for insertion into the genome. (**B**) Experimental set-up: Progeny from the transgenic lines are first screened for the expression of the red heart marker at 2 dpf. Positive larvae expressing cmlc2:*tagRFP* are heat-shocked and kept in dark conditions to prevent spontaneous Opto-ASC activation throughout the experiment. Imaging is performed at 3 dpf. (**C**) Schematic depicting the stochastic expression of Opto-ASC. Left: overview of larva; right: diagram of the epidermis with the periderm on top (dark colour) and basal cell layer below (light colour). (**D**) Time lapse images of 3dpf larva expressing Opto-ASC. Expression of Opto-ASC becomes detectable at 6 hr post heat shock. The frame rate is 15 min, time points are hours after heat shock, scale bar is 200 μm. (**E–F**) Example of Opto-ASC forming specks in the epithelial layer of 3dpf larva expressing Opto-ASC. Scale bars are 20 μm. (**E**) Time-lapse imaging after 488 nm laser illumination of the periderm cells. Top row: ASC-expressing cell forming a speck (t=8 mins). Bottom row: morphology of the dying cell in bright field. Within a minute of speck formation, the cell changes morphology and is extruded. All bright field images are at the plane of the periderm cells; fluorescent images are z-projections

*Figure 1 continued on next page*

*Figure 1 continued*

30 planes (z=1 μm). (**F**) Local activation of Opto-ASC in a single cell. Diagram of periderm showing four cells expressing Opto-ASC and region of optogenetic activation (blue square). The cell with the lowest expression of Opto-ASC was illuminated by 2-photon laser. Only this cell forms a speck.

The online version of this article includes the following figure supplement(s) for figure 1:

**Figure supplement 1.** *Control experiments for speck formation in muscle cells with Opto-ASC and with mutated Cry2olig in periderm cells.*

## Optimization of heat shock and light exposure conditions for efficient Opto-ASC speck induction

During our initial experiments, we observed variability in the efficiency of speck formation both between different larvae and between cells within a single larva. To identify conditions for efficient induction of speck formation, we titrated both the heat shock duration to maximize the expression of the construct and the laser illumination parameters to optimize the light-induced oligomerization of Cry2olig.

To first determine the optimal heat shock parameters, we crossed the Opto-ASC line to a Krt4:AKT-PH-GFP line, which marks the membranes of skin cells and allowed us to count individual cells (*Figure 2A*). We incubated the larvae at 39 °C for different periods of time and calculated the fraction of periderm cells expressing detectable levels of Opto-ASC and the overall mean fluorescence intensity (MFI) of a defined region in each larva (*Figure 2C*). While we found no significant correlation between the number of expressing cells and overall MFI in any of the heat shock schemes (*Figure 3D*), we detected high expression (>12.000 MFI) and a high percentage of expressing cells (>30 %) in those larvae that had been heat-exposed for 35 min or longer. After longer heat shocks (40 min) the larvae started to show deformations in their somites. We therefore performed all further experiments with 35 min heat shocks and, unless stated otherwise, used larvae expressing Opto-ASC in >30% of their cells for evaluation.

To determine the effect of Opto-ASC expression levels on the efficiency of speck formation, we exposed the heat-shocked larvae to 5% (1.24mWatt/μm$^2$) 488 nm laser light (an intensity normally used to image our GFP reporter lines). After 60 min, we counted the percentage of Opto-ASC-positive epithelial cells in the imaged area and the percentage of speck-forming cells among those Opto-ASC positive cells (*Figure 2E*). We found no correlation between the number of positive cells and the likelihood of speck formation. However, specks only ever formed in larvae in which more than 10% of the cells were Opto-ASC positive, indicating that there is a threshold for Opto-ASC expression below which cells do not form a speck. *Figure 2F* shows three examples of larvae with different fractions of Opto-ASC expressing cells which had been exposed to either 5% or 1% 488 nm laser light. Larvae in which few cells expressed Opto-ASC (here 12.5% of cells) rarely formed specks, even after long exposure times (1 stack/min for 60 min). Larvae with many expressing cells formed specks within less than five minutes, regardless of laser power. In larvae with high Opto-ASC levels, laser power as low as 0.1% (0.025 mWatt/μm$^2$) was sufficient to induce speck formation after several minutes (*Figure 2F*, *Video 4*).

We also observed an Opto-ASC expression level threshold for speck formation when assessing

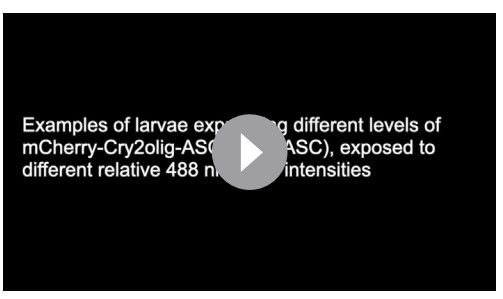

**Video 1.** Overnight time lapse of 3dpf larva expressing mCherry-Cry2olig-ASC (Opto-ASC) without blue light exposure (video sequence 1) and then being exposed to 488 nm laser light. Time is in minutes and the frame rate is 15 min. The scale bar is 200 μm.
https://elifesciences.org/articles/86373/figures#video1

**Video 2.** Example of three periderm cells of a 3 dpf larva expressing Opto-ASC that form specks after exposure to 488 nm blue light. Scale bars are 20 μm.
https://elifesciences.org/articles/86373/figures#video2

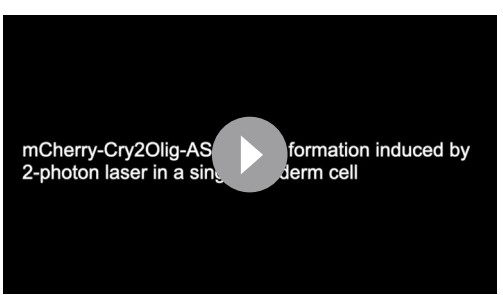

**Video 3.** Local activation of Opto-ASC (red) in a single periderm cell (white endogenous ASC-GFP visible in illuminated area and speck). Frame rate is 30 s, scale bar is 20 µm.

https://elifesciences.org/articles/86373/figures#video3

individual cells. We measured the MFI of single cells in 14 larvae and scored for each cell the time it took to form a speck under repeated blue light exposure for 60 min (*Figure 2G*). While we found no absolute threshold for speck formation, 39.6% of the cells with ASC levels of >5,000 MFI (relative units) formed specks within the period of observation, usually within the first 30 min of exposure. Cells that did not form a speck within 60 min (60.4 %) all had an MFI of <5000 MFI (relative units). When comparing cells only within individual larvae, we similarly found that cells which did not form specks also had significantly lower opto-ASC expression levels (*Figure 2H*). We also compared the effects of constant illumination and a pulsed laser exposure, where a pulse is defined as the period of imaging of a single stack (40 µm in 41 steps). We compared the effect of stimulation with one and five pulses (*Figure 2I*, *Video 5*) on the cells with similar Opto-ASC expression levels, which revealed that fewer cells formed a speck after pulsed illumination than with constant light exposure: 15.7% for five pulses (average MFI: 1999) and 17.7% after one pulse (average MFI: 2063). While fewer cells were activated than with constant illumination, the temporal dynamics of speck formation and cell death were similar to that seen after constant illumination (*Figure 2J*).

Based on all of these observations, we chose the following conditions for our further experiments: after 35 min HS we select larvae based on the percentage of expressing cells (above 20%) and expose them to 5% relative 488 nm laser intensity.

## Opto-ASC speck appearance and recruitment of endogenous ASC

We have previously shown that specks induced by Asc-mKate2 efficiently recruited endogenous ASC in the *tg(asc:asc-GFP)* line, where the open reading frame of endogenous ASC was tagged with GFP (*Kuri et al., 2017*). To test if Opto-ASC behaves in the same way as Asc-mKate2, we assessed endogenous ASC and opto-ASC co-localization in single cells in larvae expressing endogenous ASC coupled to GFP (ubiquitous expression) and Opto-ASC (mosaic expression) using Airyscan high resolution microscopy (*Figure 3A*, *Video 6*). Endogenous ASC-GFP and Opto-ASC colocalized in the cortical actin ridges, and during speck formation they co-aggregated with the same dynamics (*Figure 3—figure supplement 1A & B*). Like ASC-mKate2, Opto-ASC was present throughout the entire speck together with endogenous ASC-GFP (*Figure 3—figure supplement 1C & D*).

Opto-ASC specks were on average smaller than ASC-mKate2 specks (*Figure 3—figure supplement 1E*), and unlike ASC-mKate2, Opto-ASC occasionally formed multiple specks which often subsequently fused as the cell started to die. As an example of this, *Figure 3B* shows two neighboring cells with similar Opto-ASC expression forming specks after light exposure. Both cells recruited ASC to their membrane but while cell I formed a single speck, cell II formed multiple irregularly shaped specks (*Video 6*).

## Role of endogenous ASC and its domains in Opto-ASC-induced speck formation and cell death

All experiments reported so far were done in fish that had normal, endogenous ASC in addition to the Opto-ASC. Since Opto-ASC recruits endogenous ASC, we do not know whether cell death is triggered by endogenous ASC or Opto-ASC. To test this, we used an ASC loss-of-function mutant, $ASC^{\Delta2/\Delta2}$, in which the endogenous ASC has a CRISPR/Cas9-induced frameshift mutation that introduces a premature stop codon at Phe13 to create a protein null allele. To determine the ability of the two different ASC domains, PYD and CARD, to induce speck formation and cell death we generated Opto-PYD and Opto-CARD (both N-terminally fused to mCherry-Cry2olig). We injected full-length Opto-ASC, Opto-PYD and Opto-CARD into the $ASC^{\Delta2/\Delta2}$ mutant line. We first tested the ability of Asc-mKate2 to function in this mutant and found that it formed specks and induced cell death at the

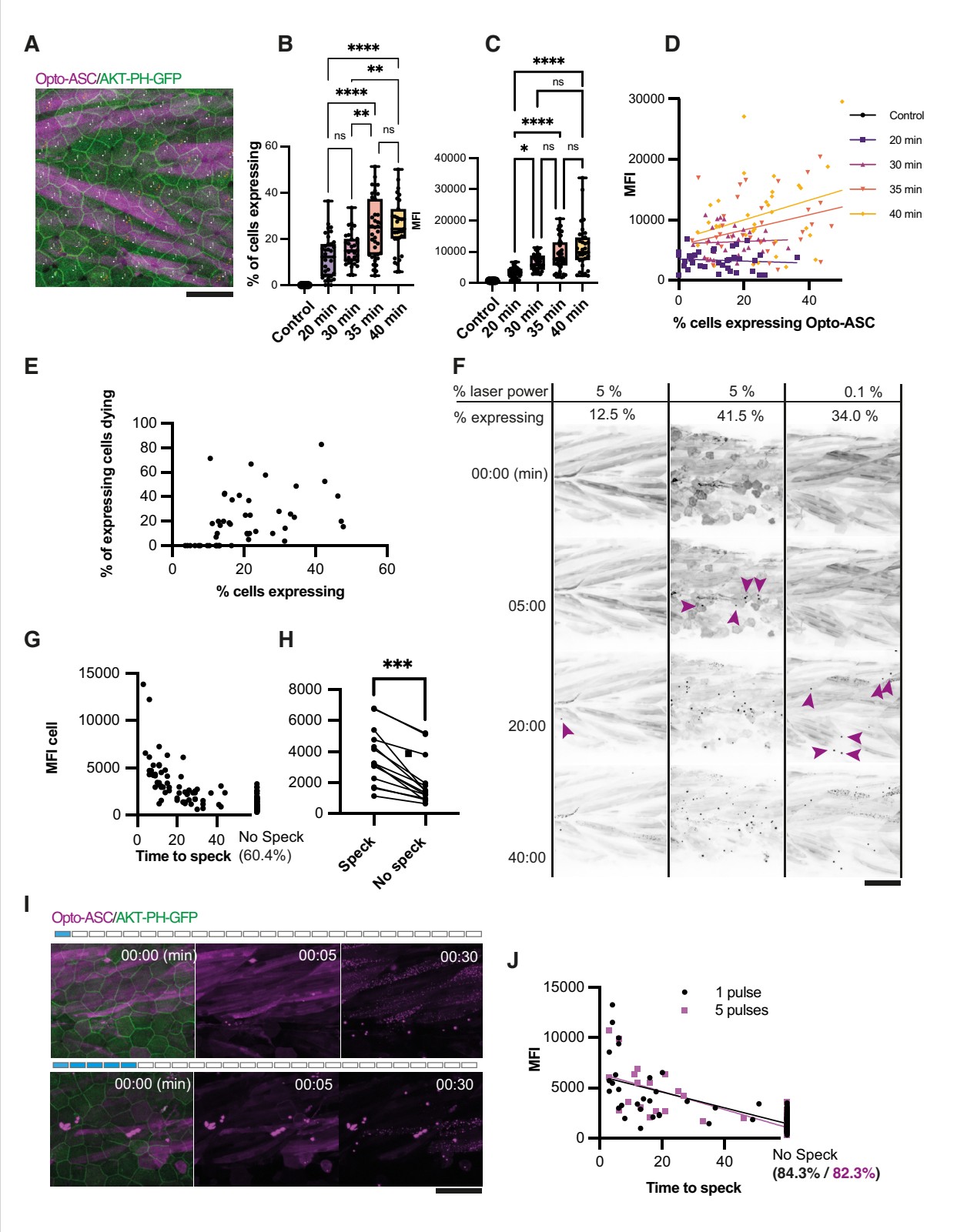

**Figure 2.** *Effect of heat shock and light exposure.* (**A**) Periderm in a larva expressing Opto-ASC (magenta) and AKT-PH-GFP to mark membranes (green) at 16 hr post heat shock (hrpHS). Cells are individually numbered for subsequent quantification of ASC expression. Opto-ASC in this figure is mainly visible in muscles underlying the epidermis, which are seen in this maximum Z-projection. For quantification of Asc-expressing periderm cells, individual z-planes within the periderm are evaluated. Scale bar: 50 µm (**B–D**) Dependence of Opto-ASC expression on heat-shock protocol. (**B**) Percentage of

*Figure 2 continued on next page*

*Figure 2 continued*

Opto-ASC expressing cells and (**C**). overall mean fluorescence intensity (MFI, measured on max projections) in larvae heat-shocked for different time periods (N>29 for each condition; each dot represents one larva). (**D**) Correlation between percentage of Opto-ASC-expressing cells and overall MFI for all larvae for each condition measured in B (N=136/>29 per condition). (**E**) Correlation between the percentage of Opto-ASC expressing cells and the number of expressing cells dying within 60 min (N=47 movies; each dot one larva); all movies at 1 min frame rate, 5% relative laser power. (**F**) Examples of larvae expressing Opto-ASC after exposure to 488 laser light at different laser powers. The number of Opto-ASC expressing periderm cells (shown as % positive cells) does not correlate strictly with the level of laser power, nor does the expression level in the underlying muscles. If a sufficient number of periderm cells are positive, specks are formed even at low laser power. First appearance of specks is indicated by magenta arrows. (**G**) Correlation between MFI of Opto-ASC in single cells and the time from beginning of 488 laser light exposure to speck formation. Cells were followed for 60 min; 60.4% of the cells had not formed a speck at this time. N=156 cells from 14 different larvae. (**H**) Comparison of average MFI between Opto-ASC expressing cells in individual larvae that form and do not form a speck within 60 min (N=15 individual larvae). The values representing the average MFI for speck-forming and non-speck-forming cells for each larva are connected by a line; p<0.001. (**I**) Time lapse of Opto-ASC expressing larva exposed to a single or to 5 laser light pulses (1 or 5 frames acquired with 488 laser light). Scale bar is 50 μm. (**J**) Correlation between MFI of single cells and onset of speck formation in cells exposed to either 1 (red circles) or 5 (black outlined squares) pulses of 488 laser light. Cells were followed for 60 min. N=164 cells in 9 larvae for 1 pulse, N=86 cells in 5 larvae for 5 pulses.

same level as in the wild-type larvae, showing that Asc-mKate2 is fully functional and able to induce cell death (*Figure 4A*, *Video 7*).

In the wild-type ASC-GFP line, Opto-Asc, Opto-PYD and Opto-CARD induced speck formation within five minutes, whereas in the ASC$^{Δ2/Δ2}$ line we observed delayed speck formation (>2.5 min after first aggregates were observed (N=3)). For all of the constructs, the shrinking of the cell, which we consider an initial sign of cell death upon Opto-ASC activation, was seen as soon as the first oligomerization was detectable.

By contrast, in the ASC$^{Δ2/Δ2}$ knockout line, Opto-ASC, Opto-PYD and Opto-CARD formed multiple smaller aggregates rather than a single large speck, indicating that endogenous ASC may facilitate

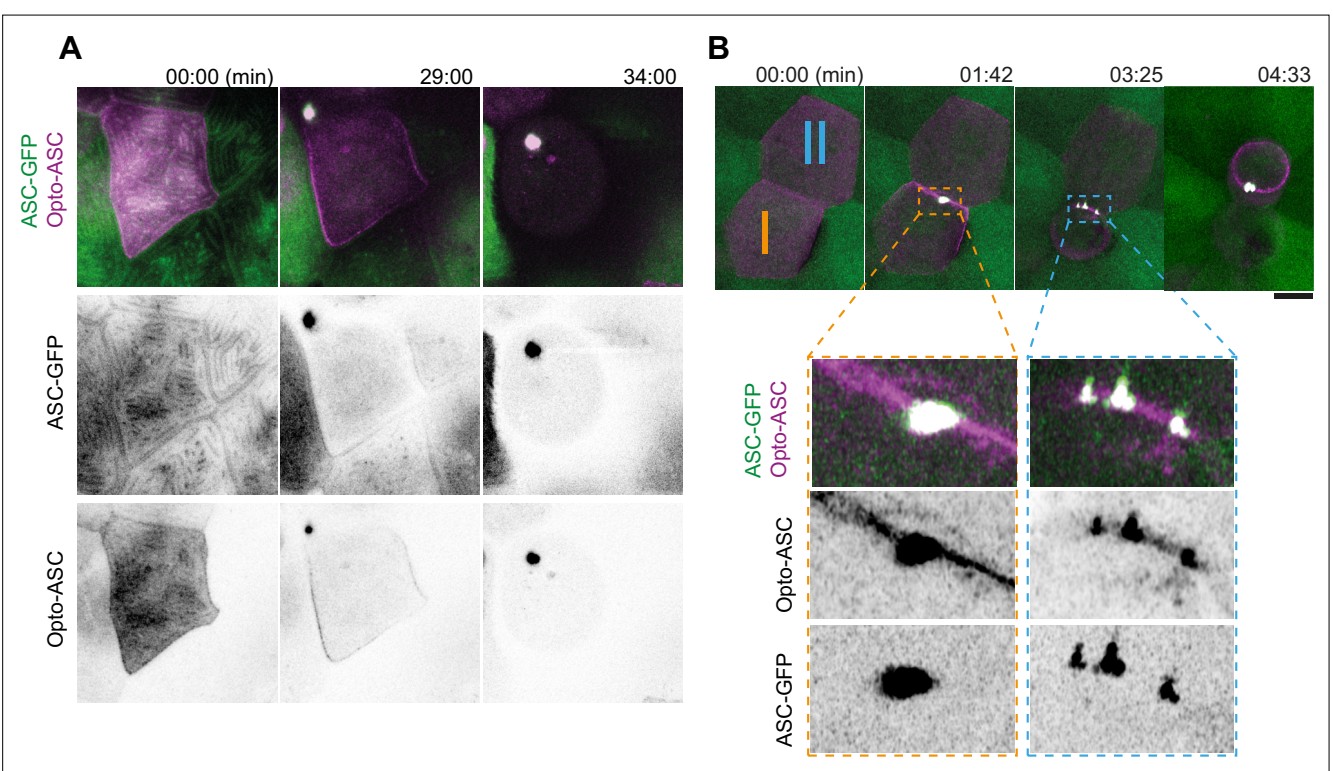

**Figure 3.** Speck formation dynamics of Opto-ASC and recruitment of endogenous ASC. (**A**) Time lapse of speck formation induced by Opto-ASC (magenta) and recruitment of endogenous ASC (green, ubiquitous) in a periderm cells. Scale is 10 μm. (**B**) Two neighboring cells (I and II) forming either a single speck (cell I) or multiple specks (cell II) along the cell membrane. The specks coalesce as the cell shrinks. Scale is 2 μm.

The online version of this article includes the following figure supplement(s) for figure 3:

**Figure supplement 1.** Recruitment dynamics in Opto-ASC specks and ASC-mKate2 specks.

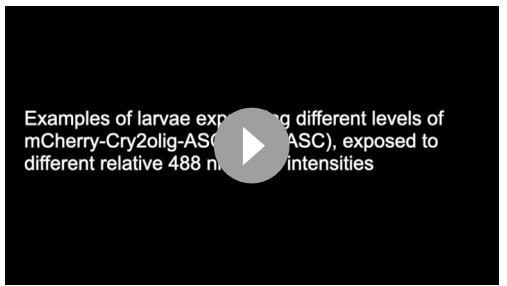

**Video 4.** Three examples of larvae expressing Opto-ASC (magenta) exposed to 488 nm laser light at different laser powers. Membranes are marked with AKT-PH-GFP (green). Time is in minutes, scale bar is 20 μm.

https://elifesciences.org/articles/86373/figures#video4

**Video 6.** High resolution imaging of speck formation in a single (first video sequence) and two neighbouring periderm cells (second video sequence). Opto-ASC (magenta) speck formation is induced by 488 nm laser light illumination. Endogenous ASC (green, ubiquitous) is recruited to the speck. Time is in minutes.

https://elifesciences.org/articles/86373/figures#video6

multiple Opto-ASC aggregates to coalesce to form a single speck. In this mutant, Opto-CARD did not induce cell death, consistent with our earlier finding that inflammatory caspases are recruited via the PYD of ASC (*Kuri et al., 2017*), and therefore, without endogenous ASC, caspases could not be recruited to the Opto-CARD construct. By contrast, both Opto-ASC and Opto-PYD induced cell death, but often with a delay when compared to ASC wild type cells. (*Figure 4B*, *Video 8*).

When we measured the time between the first detectable oligomerization and first visible signs of cell death, we found that in some $ASC^{\Delta2/\Delta2}$ periderm cells, death was significantly delayed while in other $ASC^{\Delta2/\Delta2}$ and wild type periderm cells, death was immediate after the first oligomers were detected (*Figure 4C*, *Video 9*). These observations indicate that the assembly of ASC into a clearly defined single speck was not required for downstream effectors to induce cell death. *Figure 4D* shows the example of three cells expressing different levels of Opto-PYD, which are normalized relative to the cell with the weakest expression (1 x). The cell with the highest level of Opto-PYD (13.5 x) entered cell death before oligomerization was detectable, whereas in cells with intermediate (4.5 x) or low (1 x) levels, Opto-PYD recruitment to the membrane and oligomerization clearly preceded the appearance of first signs of cell death. This shows that the efficiency of cell death induction depends on the amount of Opto-ASC or Opto-PYD, and that with high concentrations of Opto-ASC or Opto-PYD, the formation of a visible speck is not required for initiation of cell death.

## Cell extrusion after Opto-ASC speck formation

Periderm cells typically respond to inflammasome formation by rapid cell death and apical extrusion (*Kuri et al., 2017*). We took advantage of Opto-ASC-mediated inflammasome formation to characterize the cellular events immediately following speck formation. *Figure 5A* shows an example of a cell that was extruded from the periderm after speck formation. The cell began to shrink within 30 s of speck formation while the surrounding cells moved in to fill the space left by the dying cell (*Video 10*). The dying cell lost its shape and formed membrane blebs, eventually swelled and was finally extruded from the cell layer. Once completely extruded, the cell appeared round and lysed within minutes, as determined by the uptake of the membrane-impermeable DNA-binding dye DRAQ7 which can only stain the cells following membrane permeabilization (*Figure 5B*, *Video 11*). Most cells reached the lytic stage 5–10 min after illumination, although some remained fully extruded without lysing for up to 20 min (*Figure 5C*).

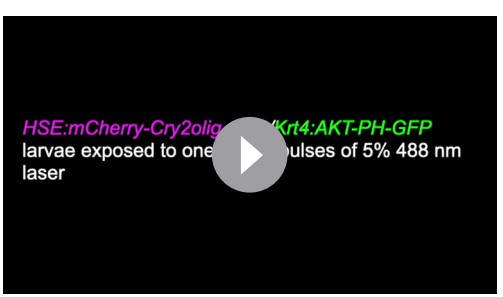

**Video 5.** Time lapse of Opto-ASC expressing larvae exposed to one or five 488 nm laser light pulses (1 or 5 frames acquired with 488 laser light). Scale bar is 20 μm. Time is in minutes.

https://elifesciences.org/articles/86373/figures#video5

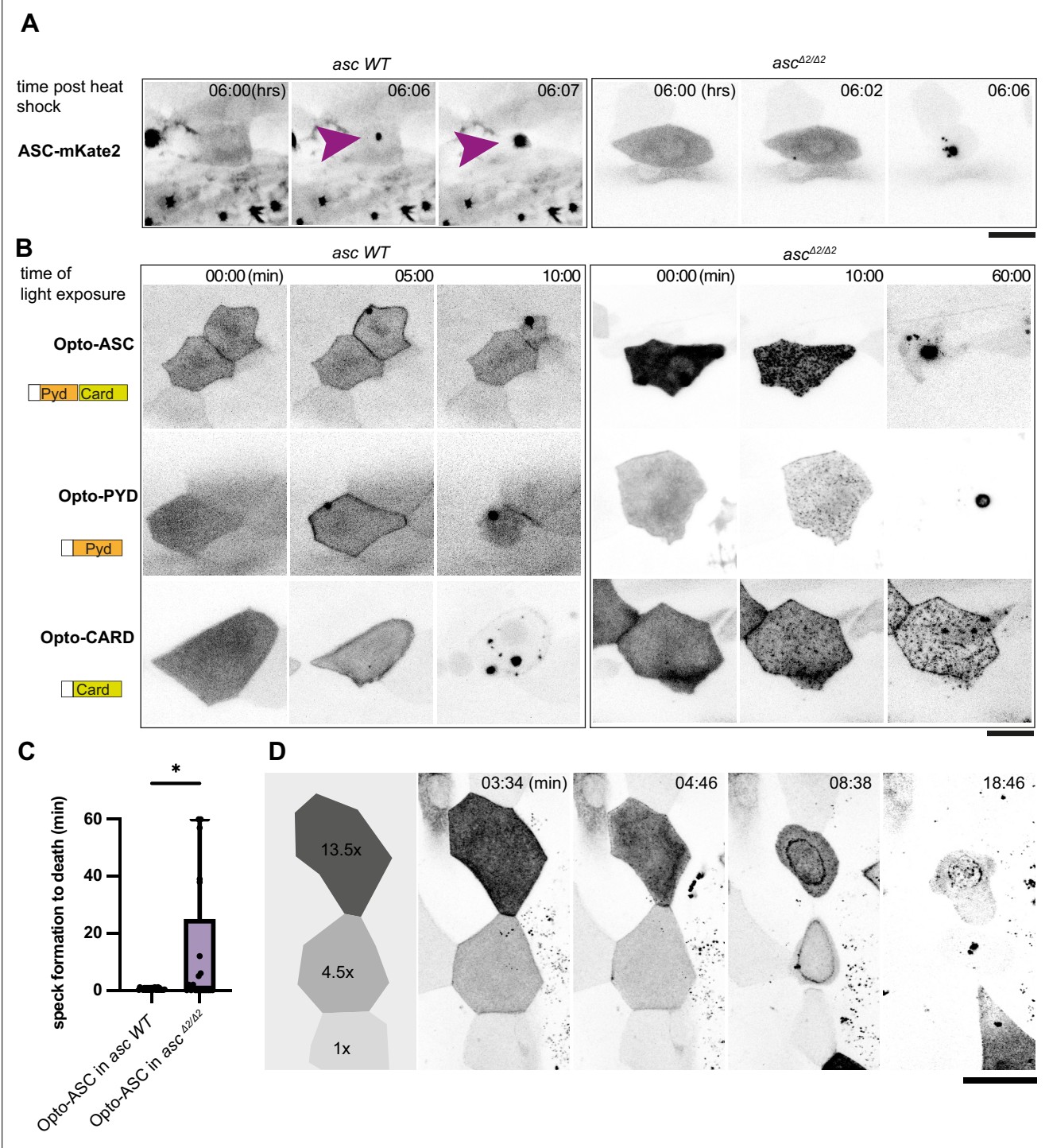

**Figure 4.** Opto-ASC speck formation and dependence on endogenous ASC. (**A, B**) Speck formation induced by ASC-mKate2, Opto-ASC, Opto-PYD and Opto-CARD in wildtype larvae and $asc^{\Delta2/\Delta2}$ larvae. Scale bar is 20 μm. (**A**) Time for ASC-induced specks is counted from end of heat shock. Purple arrow heads mark the speck in the cell of interest. Specks formed by surrounding cells can be seen as black spots. (**B**) Time is in min counted from the last timepoint before visible speck formation. (**C**) Time is measured between first detectable appearance of a speck and first morphological sign of cell death (deformation of cell) in wildtype (N=23) and $asc^{\Delta2/\Delta2}$ (N=14) larvae. *=$p < 0.05$. (**D**) Dynamics of speck formation in three cells with different expression levels of Opto-PYD. Relative intensity levels shown in the diagram on the left are measured in relation to the cell with the lowest level (1 x) after background subtraction. Time is measured in minutes; scale bar is 20 μm.

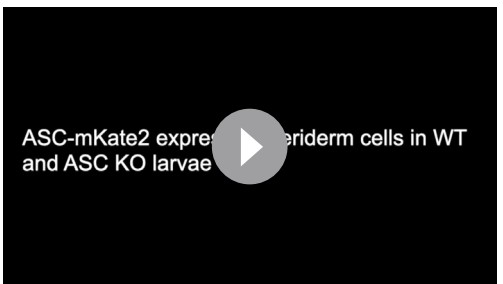

**Video 7.** Time lapse of ASC-mKate2-induced speck formation in periderm cells of wildtype and *asc^{Δ2/Δ2}* larvae. Time is in minutes.

https://elifesciences.org/articles/86373/figures#video7

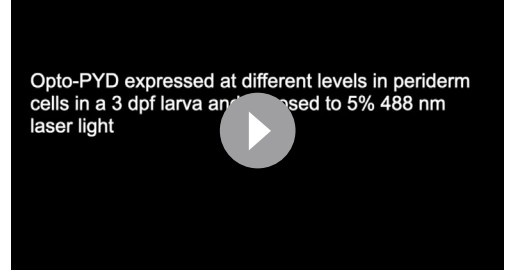

**Video 9.** Time lapse of three neighboring periderm cells in which Opto-PYD is expressed at different levels. Time is in minutes.

https://elifesciences.org/articles/86373/figures#video9

Basal cells also died following Opto-ASC speck formation, but with a different morphology. They showed extensive fragmentations with the formation of membrane vesicles labeled by Akt-PH-GFP (*Figure 5D*, *Video 12*) or actin labeled by Utr-mNeonGreen (*Figure 6—figure supplement 3*). The basal cells responded more slowly to speck formation than periderm cells and started to change morphology only after ~5 min (*Figure 5E*).

## Apical and basal cell extrusion in response to Opto-ASC specks

The fate of the dying cells following Opto-ASC speck formation varied between cells within and between different larvae. Some periderm cells were not extruded apically, as described above, but instead left the periderm on the basal side. Another subset of cells fell in neither of these groups; instead, part of their cell body was extruded apically, rounded up and lysed, whereas another part was extruded basally, morphologically resembling the basally extruded cells. As an example of this, *Figure 6A* shows cells located near each other in the same larva, responding by apical extrusion (A), basal extrusion (B) or mixed extrusion (M) (*Video 13*).

To see if this variation in phenotype was determined by genetic differences, we injected the Opto-ASC construct into different commonly used laboratory strains (*Figure 6B*). In most strains (AB, AB2B2 and Golden) basal extrusion of periderm cells only occurred in a small number (<5 %) of cells per larva, and not in all larvae. However, the WIK (Wild Indian karyotype) strain showed greater variation, as did our experimental line (*Tg(Opto-asc)* x *Tg(Krt4:AKT-PH)*) of mixed genetic background (Opto-ASC: Golden and AB2B2, Krt4:AKT-PH: unknown). In these two lines, higher percentages of dying cells were basally extruded (38% in WIK and 64% in *Tg(Opto-asc)* x *Tg(Krt4:AKT-PH)*).

To characterize the morphological differences between apical and basal cell extrusion, we analyzed the dynamics of the actin cytoskeleton in the dying cells. We visualized actin with an mNeonGreen (mNG)-tagged variant of the calponin homology domain of utrophin (UtrCH) expressed under control of the UAS-promoter (*Tg(UAS:mNG-UtrCH)*), which we combined with a Krt4:Gal4 driver, and UAS:lyn-tagRFP to label plasma membranes). Larvae carrying these constructs showed stochastic expression of mNeonGreen-UtrCH in periderm cells, which allowed us to follow actin dynamics. The overall response of the actin cytoskeleton was similar in apically (*Figure 6C*) and basally (*Figure 6D*) extruded periderm cells (*Video 14*). As the cell started to shrink after speck formation, the actin ridges rapidly disappeared. Actin then accumulated in the cortex of the cell just prior to extrusion. Basally extruded cells were fragmented once fully internalized and were taken up by surrounding cells or macrophages (*Figure 6—figure supplement 1*).

We previously showed that periderm cells dying by Opto-caspase-8-induced apoptosis were

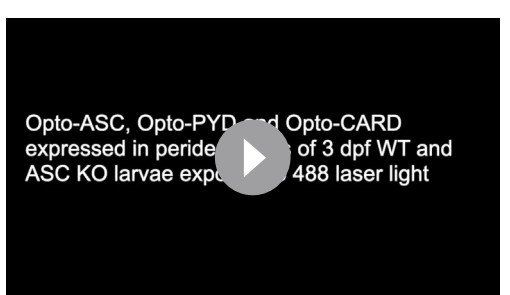

**Video 8.** Time lapse of periderm cells of larvae expressing Opto-ASC (first video sequence), Opto-PYD (second video sequence) and Opto-CARD (third video sequence) in wildtype larvae and *asc^{Δ2/Δ2}* larvae exposed to 488 nm laser light. Time is in minutes.

https://elifesciences.org/articles/86373/figures#video8

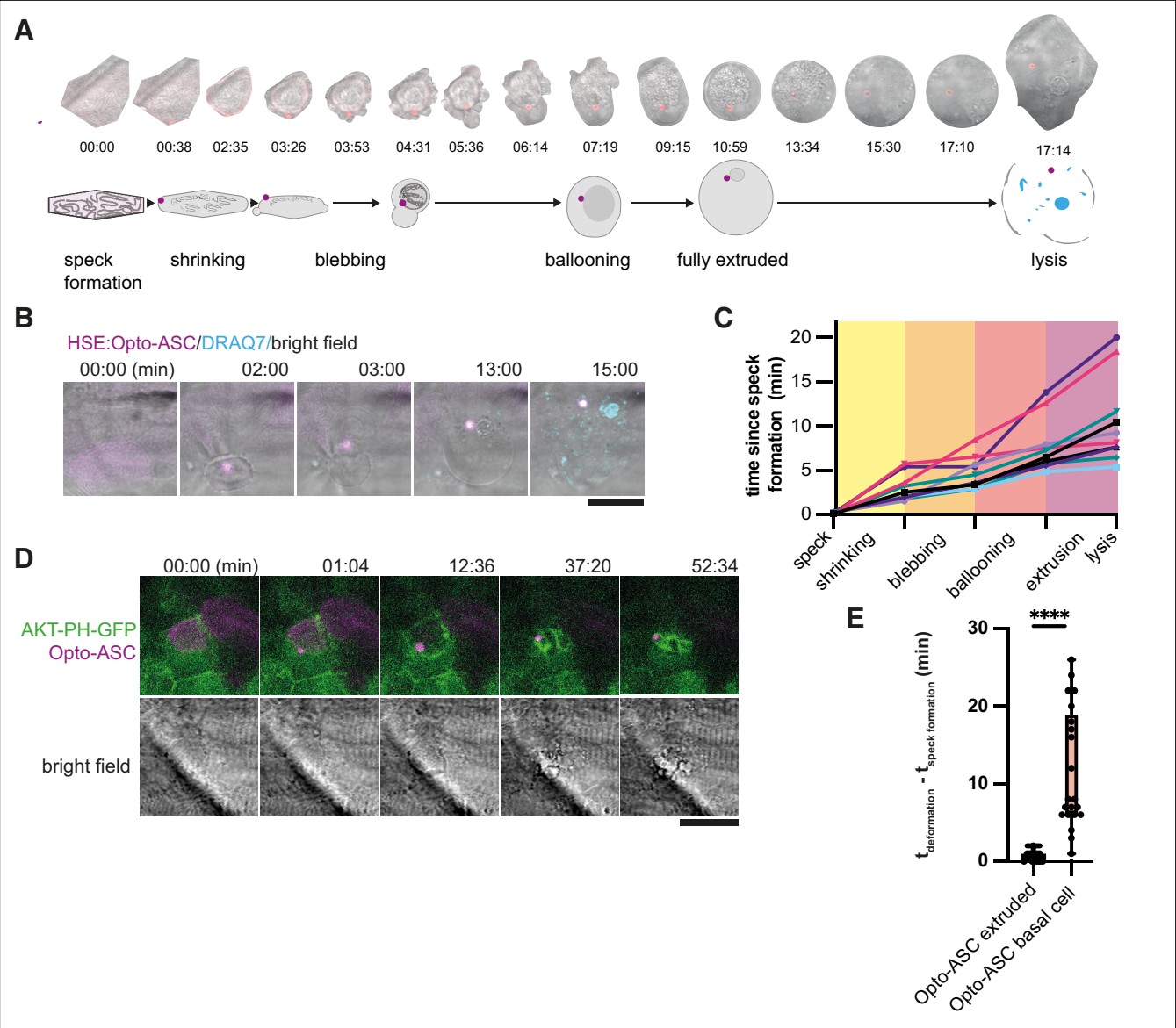

**Figure 5.** Stages of ASC-speck-induced e extrusion from the periderm. (**A**) Example of a single cell from a bright-field movie (top) and a schematic showing stages of periderm cell extrusion and lysis after speck formation (bottom). (**B**) DRAQ7 staining of a cell that lyses after the full extrusion from the periderm. Scale bar is 20 μm. (**C**) Time lines for 10 periderm cells undergoing the steps of cell extrusion described in A. Time is in minutes. (**D**) Basal cell dying after formation of an Opto-ASC-speck (magenta). Cell membrane is labeled by AKT-PH-GFP (green). (**E**) Time between Opto-ASC speck formation and the appearance of first morphological signs of cell death (deformation of cell) in periderm (N=49) and basal cells (N=20). ****=p < 0.0001.

also basally extruded from the periderm (*Shkarina et al., 2022*). We analysed the behaviour of actin in larvae derived from a cross of *tg(Opto-caspase-8)* with *tg(Krt4:Gal4_UAS:mNG-UtrCH-UAS:Lyn-tagRFP)* and found that the apical actin ridges responded differently. In contrast to the Opto-ASC-activated cells, apoptotic periderm cells retained the original pattern of actin ridges while the cell was shrinking, with a proportional scaling of the pattern until the cell was fully internalized (approximately 10 min; *Figure 6E*, *Video 14*).

In the periderm cells surrounding dying cells either after formation of an ASC speck or induction of apoptosis by Opto-caspase-8, actin was always recruited to the membrane region that was in contact with the dying cell, independent of the cell death phenotype (*Figure 6—figure supplement 2*). We also analyzed the actin cytoskeleton in basal cells dying in response to ASC-speck formation, which revealed that while the cell was shrinking, its actin was re-localized to the membrane along the borders

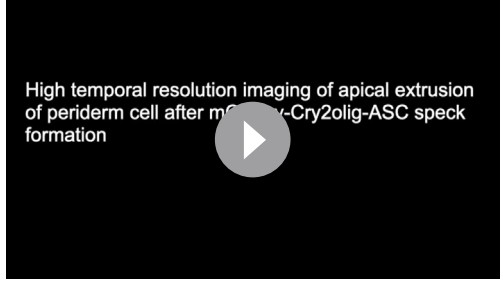

**Video 10.** Time lapse of an apically extruded cell after Opto-ASC induced speck formation at high temporal resolution. Frame rate is 13 s.
https://elifesciences.org/articles/86373/figures#video10

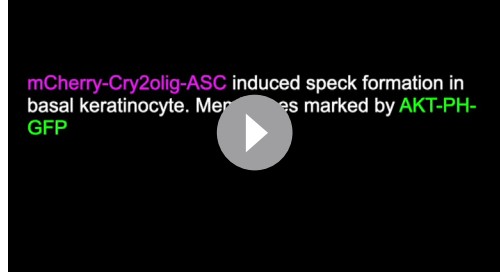

**Video 12.** Basal cell dying after formation of an Opto-ASC-speck (magenta). Cell membrane is labeled by AKT-PH-GFP (green).
https://elifesciences.org/articles/86373/figures#video12

to neighboring cells (*Figure 6—figure supplement 3*, compare A vs. B) at the same time as those cells started to migrate to close the gap created by the dying cell (*Figure 6—figure supplement 3B*).

## Role of caspases in Opto-ASC-induced apical extrusion

Opto-ASC efficiently induced cell death in periderm and basal cells, but not in muscle cells which lack the necessary downstream effector caspases (*Spead et al., 2018*). To test the role of inflammatory caspases in speck-induced cell death, we disrupted the function of the two caspase-1 homologues, *caspa* and *caspb,* by outcrossing *Tg(Opto-asc)* to a *caspa* knock-out (KO) line (*Kuri et al., 2017*), but unexpectedly saw no effect on the extrusion of periderm cells after Opto-ASC speck formation. We also used a CRISPR/Cas9 based method to create F0 knock-out animals, a method that efficiently creates protein-null alleles (*Kroll et al., 2021*), as an alternative strategy of interfering with caspase expression. The synthetic short guide RNAs (sgRNAs) targeting different exons of *caspa* and *caspb* labeled in red in *Figure 7A* were used in the double KO, and the ones labelled in grey for single gene KOs. The efficiency of the knock-out was validated by sequencing around all sgRNA binding sites.

*caspa* deletion had no effect on the morphology or timing of Opto-ASC-induced cell death or the extrusion efficiency (around 90% of cells were still apically extruded). However, periderm cells lacking Caspb were no longer extruded apically (*Figure 7B and C*, *Video 15*), but instead they displayed the apoptotic phenotype, similar to the one observed in these cells after Opto-caspase-8 activation (*Figure 7D*, *Video 16*). In particular, these cells shrank while retaining the pattern of actin ridges (*Figure 7E*). Thus, either Caspa or Caspb alone is sufficient to trigger cell death, but only Caspb can trigger apical extrusion.

In mammalian cells, inflammasome formation results in pyroptosis, which is defined as cell death induced by Gasdermin pore formation (*Liu et al., 2016*). The zebrafish genome encodes two Gasdermins (Gsdms): GsdmEa and GsdmEb, which can be cleaved in vitro by Caspb and apoptotic caspase-8 and –3 (*Wang et al., 2020*; *Chen et al., 2021*). To test the role of these Gsdms in Opto-ASC-induced cell death and extrusion, we used two ways of inactivating Gsdm function. First, we made F0 CRISPR/Cas9 knock-outs with four synthetic sgRNAs for each geneimental procedures were app. We imaged and genotyped at least five larvae for each of the single and double knockouts of the Gsdms. We also used a range of Gsdm inhibitors. LDC7559to inhibit Gsdm pores of neutrophils (*Isles et al., 2021*), although it may instead act through a different pathway via PFKL (*Amara et al., 2021*). Disulfiram and dimethyl fumarate (DMF) have been shown to inhibit Gasdermin D pore formation and thereby prevent pyroptosis (*Figure 7—figure supplement 1*). None of these drugs had any effect on the phenotype or dynamics of cell death or on the polarity of extrusion (*Figure 7—figure supplement 2*).

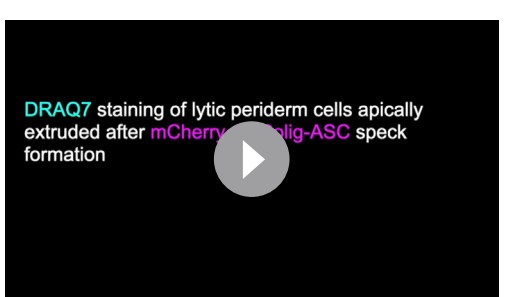

**Video 11.** DRAQ7 (cyan) staining of a cell that lyses after its extrusion from the periderm in response to Opto-ASC-induced speck formation (magenta).
https://elifesciences.org/articles/86373/figures#video11

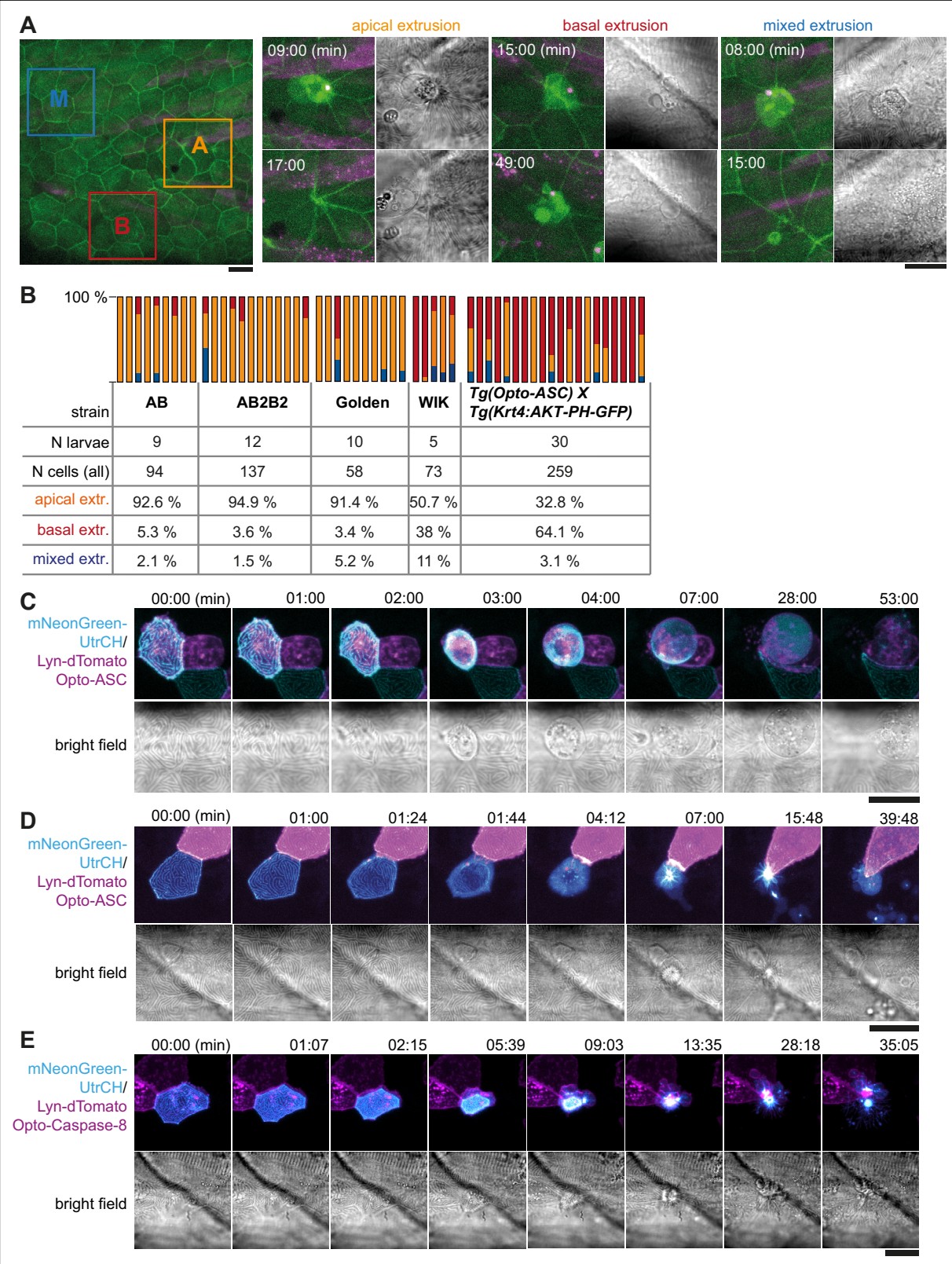

**Figure 6.** Apical and basal extrusion during Opto-ASC induced cell death. (**A**) Examples of responses to Opto-ASC (magenta) speck formation in three periderm cells within the same larva (marked by blue, orange and red squares in the overview image); cell borders marked by AKT-PH-GFP, green. Dying periderm cells can be extruded apically (orange), basally (blue) or in both directions (red). (**B**) Percentage of periderm cells extruded either apically, basally or in both directions (mixed) after Opto-ASC speck formation in different *Danio rerio* laboratory strains and transgenic *Tg(Opto-asc X Krt4:Akt-*

*Figure 6 continued*

*PH-GFP*) larvae. Each bar represents one larva, the y-axis shows what fraction of cells undergoes which type of extrusion. (**C–D**) Response of the actin cytoskeleton in cells that are apically (**C**) or basally (**D**) extruded after formation of an ASC speck (magenta) shown as z-projections. Actin is labeled using mosaic expression of mNeonGreen-UtrCH (cyan). The apical cortical actin ridges can also be seen in bright field images. Membranes of cells are mosaically labeled by expression of lyn-tagRFP (also magenta). (**E**) Actin response to Opto-Caspase-8-induced apoptosis in periderm cells; actin in the dying cell is in cyan (mNeonGreen-UtrCH). Membranes of cells are mosaically labeled by lyn-tagRFP (also magenta). Scale bars in all images are 20 μm.

The online version of this article includes the following figure supplement(s) for figure 6:

**Figure supplement 1.** Response of neighbouring cells to cell death after ASC-speck formation.

**Figure supplement 2.** Actin rearrangement in periderm cells near dying cells.

**Figure supplement 3.** Actin dynamics in neighboring cells after Opto-ASC-induced cell death in basal cells.

We also measured the time span between speck formation and first signs of cell death, in this case the start of cell deformation (shrinking) in different F0 KO backgrounds. F0 KO of *caspb* led to a delay in the response to speck formation of up to 20 min, regardless of the direction of extrusion (*Figure 7F*). Double deletion of *caspa* and *caspb* delayed Opto-ASC-induced cell death up to 180 min or longer. Similar to cells in which apoptosis was induced by Opto-caspase-8, these dying cells required a longer time (around 15 min) to be extruded from the periderm than during Opto-ASC induced cell death in control larvae (*Figure 7G*).

We also observed differences in the number and size of cell fragments, as measured by the number and diameter of Akt-PH-GFP labeled membrane vesicles in basally extruded periderm cells, depending on whether death was induced by Opto-ASC with or without Caspb or by Opto-caspase-8 (*Figure 7H*). Again, the phenotype of Opto-ASC-induced death in the absence of Caspb resembled that of Opto-caspase-8-induced apoptosis. We will therefore refer to this phenotype as Opto-ASC-induced apoptosis.

To test if either caspase-8 or caspase-3 was involved in the execution of ASC-induced cell death we knocked out both genes using the CRISPR/Cas9-based method with four synthetic sgRNAs. However, in both cases the loss of the gene resulted in embryonic lethality so that effects at the larval stage could not be assessed.

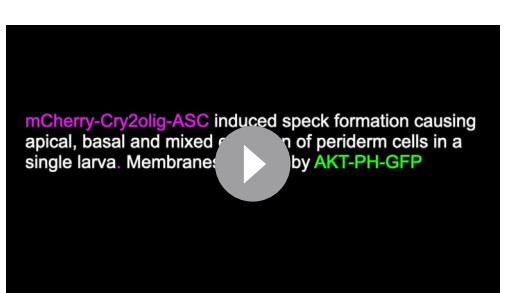

**Video 13.** Thee examples of responses to Opto-ASC (magenta) speck formation in three periderm cells within the same larva (marked by blue, orange and red squares in the overview image); cell borders marked by AKT-PH-GFP, green. Overview image (movie sequence 1). Apically extruded cell (image sequence 2), basally extruded cell (image sequence 3) or periderm cell extruded in both directions (image sequence 4). ree examples of responses to Opto-ASC (magenta) speck formation in three periderm cells within the same larva (marked by blue, orange and red squares in the overview image); cell borders marked by AKT-PH-GFP, green. Overview image (movie sequence 1). Apically extruded cell (image sequence 2), basally extruded cell (image sequence 3) or periderm cell extruded in both directions (image sequence 4).

https://elifesciences.org/articles/86373/figures#video13

## Ca²⁺ signaling in response to different types of cell death

Pyroptotic cells have been shown to attract phagocytic cells via the release of ATP and other signaling molecules (*Wang et al., 2013*). To see if dying cells affect surrounding cells we looked

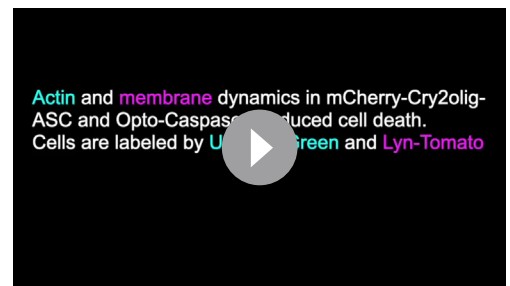

**Video 14.** Response of the actin cytoskeleton in cells that are apically (image sequence 1) or basally (image sequence 2) extruded after formation of an ASC speck (magenta) or dies apoptotic after Opto-Caspase-8 activation (image sequence 3). Actin is labeled using mosaic expression of mNeonGreen-UtrCH (cyan). Membranes of cells are mosaically labeled by expression of lyn-tagRFP (also magenta).

https://elifesciences.org/articles/86373/figures#video14

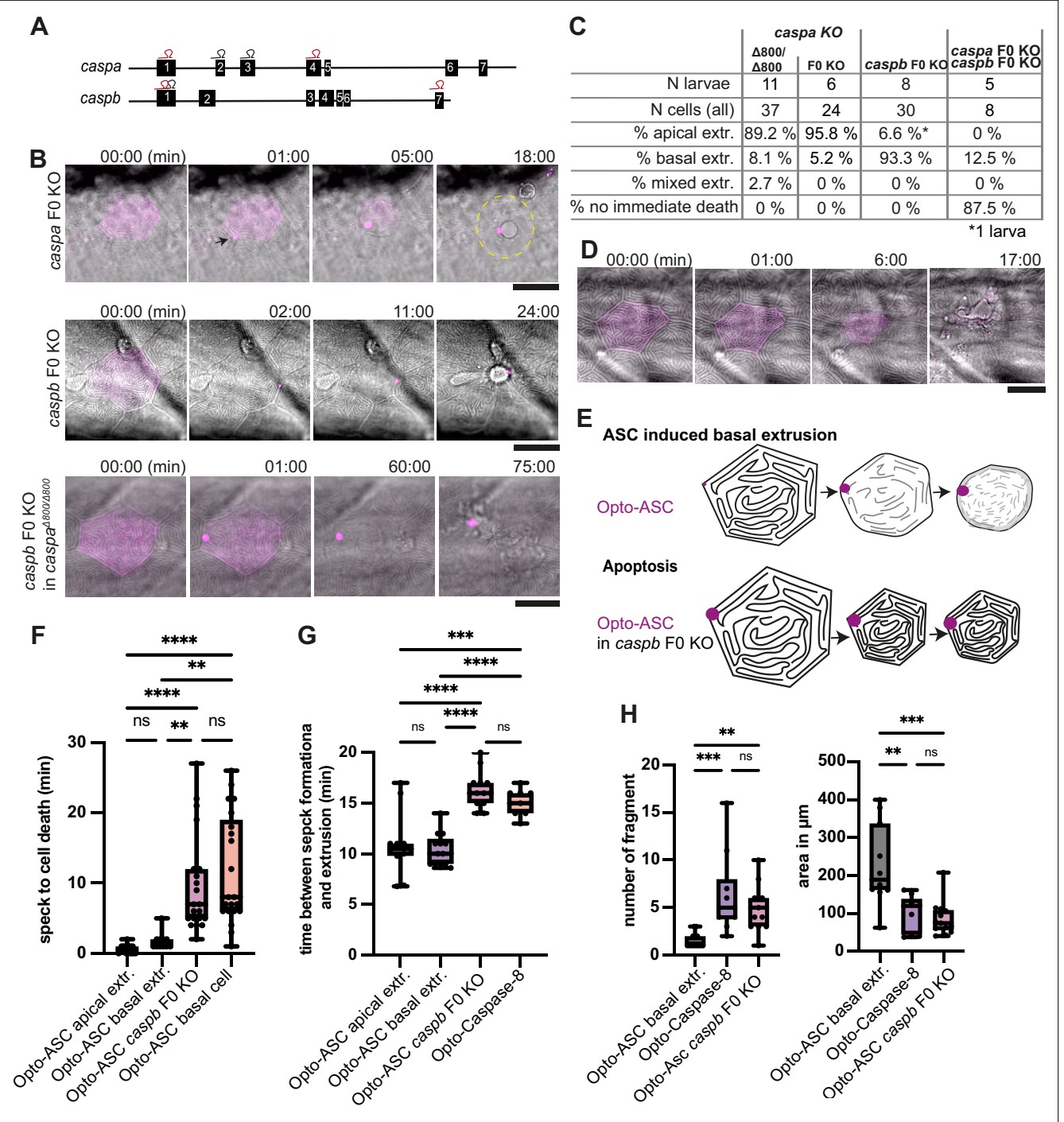

**Figure 7.** Inflammatory caspases in cell extrusion and rapid cell death after ASC-speck formation. (**A**) Schematic of the *caspa* and *caspb* genes and sgRNA binding sites for F0 CRISPR/Cas9 knock-outs (F0 KO). (**B**) Representative responses of periderm cells after speck formation in *caspa* F0 KO, *caspb* F0 KO and *caspa* F0 KO/*caspb* F0 KO. Scale bar is 20 µm. (**C**) Percentages of apically extruded and basally extruded cells and cells that do not die within the observation time of >60 min after Opto-ASC speck formation in different F0 KO backgrounds. (**D**) Periderm cell dying by apoptosis after Opto-Caspase-8 induction; scale bar is 20 µm. (**E**) Schematic of basally extruding cell after Opto-ASC induction and isomorphic shrinkage as seen in in *caspb* F0 KO larvae after Opto-ASC induction. (**F**) Time between Opto-ASC speck formation and first morphological signs of cell death (deformation of cell) in the indicated conditions. (**G**) Time of gap closure after death of single periderm cells which are either extruded or retained after the indicated treatments. (**H**) Number of fragments at the time of gap closure and area of the largest cell fragment of retained cells after ASC-Speck formation in WT and *caspb* F0 KO larvae and of apoptotic cells after Opto-Caspase-8 induction. (**F–H**) **=p < 0.01, ***=p < 0.001, ****=p < 0.0001.

The online version of this article includes the following figure supplement(s) for figure 7:

*Figure 7 continued on next page*

*Figure 7 continued*

**Figure supplement 1.** Extrusion of periderm cells after ASC speck formation under treatment with Gasdermin D or pyroptosis inhibitors.

**Figure supplement 2.** Transient expression of HSE: GFP/ASC-mKate2 in basal cells.

at Ca²⁺ signaling in the zebrafish skin. We used the Ca²⁺ signaling reporter GCamp6 to characterize the response of surrounding cells to cell death induced by Opto-ASC specks or Opto-caspase-8. We crossed the *Tg(Opto-asc)* line to *Tg(ß-actin:GCamp6)*, a line expressing GCamp6 under the ubiquitously active ß-actin promoter (*Chen et al., 2017*) and imaged single cells expressing Opto-ASC and their immediate surrounding at high temporal resolution (<5 s/frame).

Without induction of cell death, only sporadic and weak Ca²⁺ signals were seen in individual cells at a frequency of about 0.2 per minute (3 cells in 15 min) within an imaging area of around 24 cells corresponding to 13 mm² (*Figure 8—figure supplement 1A*). In contrast, Opto-ASC activation induced strong Ca²⁺ responses in neighbours. As an example, *Figure 8A* shows Ca²⁺ signaling in the periderm surrounding a cell being apically extruded following Opto-ASC speck formation (*Video 17*). The speck-forming cell as well as the basal cell underlying the site of the speck showed increased Ca²⁺ levels within 15 s of the speck becoming detectable; this preceded the shrinking of the cell, which started after 30 s. Ca²⁺ signaling then spread to the surrounding cells in a wave-like manner. The first intense Ca²⁺ wave was immediately followed by a smaller second wave, after which the signal became sporadic over the course of 5 min and then subsided. Once the dying cell started blebbing, its internal Ca²⁺ level remained high until lysis, indicating the loss of its ability to control intracellular Ca²⁺ homeostasis.

To analyze these observations in a more quantitative manner, we used a two-dimensional representation of time and space (derived from 3D kymographs; see *Figure 8—figure supplement 1*) to represent the calcium response in the area around the dying cell (*Figure 8B*, *Video 17*). The sum of the signals along a radial sweep around the cell is represented on the y-axis (with the center of the dying cell at y=0) versus time on the x-axis (*Figure 8—figure supplement 1B & C*). The initial Ca²⁺ waves (in cyan) appear as strong lines following the appearance of the speck (yellow arrow), and the sporadic Ca²⁺ signals of surrounding cells appear as the weaker lines. The speck is visible following its appearance as a magenta line, until it disappears following cell lysis. The extruded cell appears as a cyan shadow once extruded, and remains visible until lysis.

Cells that were basally extruded responded to speck formation with immediate increased Ca²⁺ levels in three out of five cases. To estimate the response of cells in the vicinity of the dying cell we counted the number of cells with increased Ca²⁺ levels in the first 15 min after the cell started to shrink. For apically extruded cells we detected on average 9 basal cells and 31 periderm cells with increased Ca²⁺ levels forming an Opto-ASC speN=3 dying cells for each condition (*Figure 8—figure supplement 2A, B*).

By contrast, caspase-8-induced apoptosis of periderm cells did not induce a comparable Ca²⁺ wave in the surrounding periderm (*Figure 8D*, *Video 18*), and apoptotic cells themselves never showed increased Ca²⁺ levels. During the shrinking phase, before the cell was fully internalized, the surrounding cells (mainly basal cells) showed only

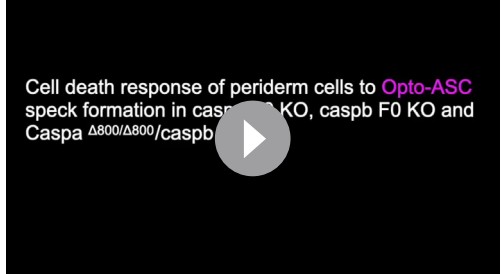

**Video 15.** Representative responses of periderm cells after speck formation in *caspa* F0 KO (image sequence 1), *caspb* F0 KO (image sequence 2) and *caspa* F0 KO/*caspb* F0 KO (image sequence 3). Scale bar is 20 µm.

https://elifesciences.org/articles/86373/figures#video15

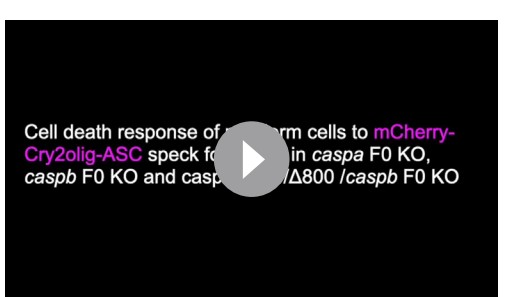

**Video 16.** Periderm cell dying by apoptosis after Opto-Caspase-8 induction; scale bar is 20 µm. Time is in hours.

https://elifesciences.org/articles/86373/figures#video16

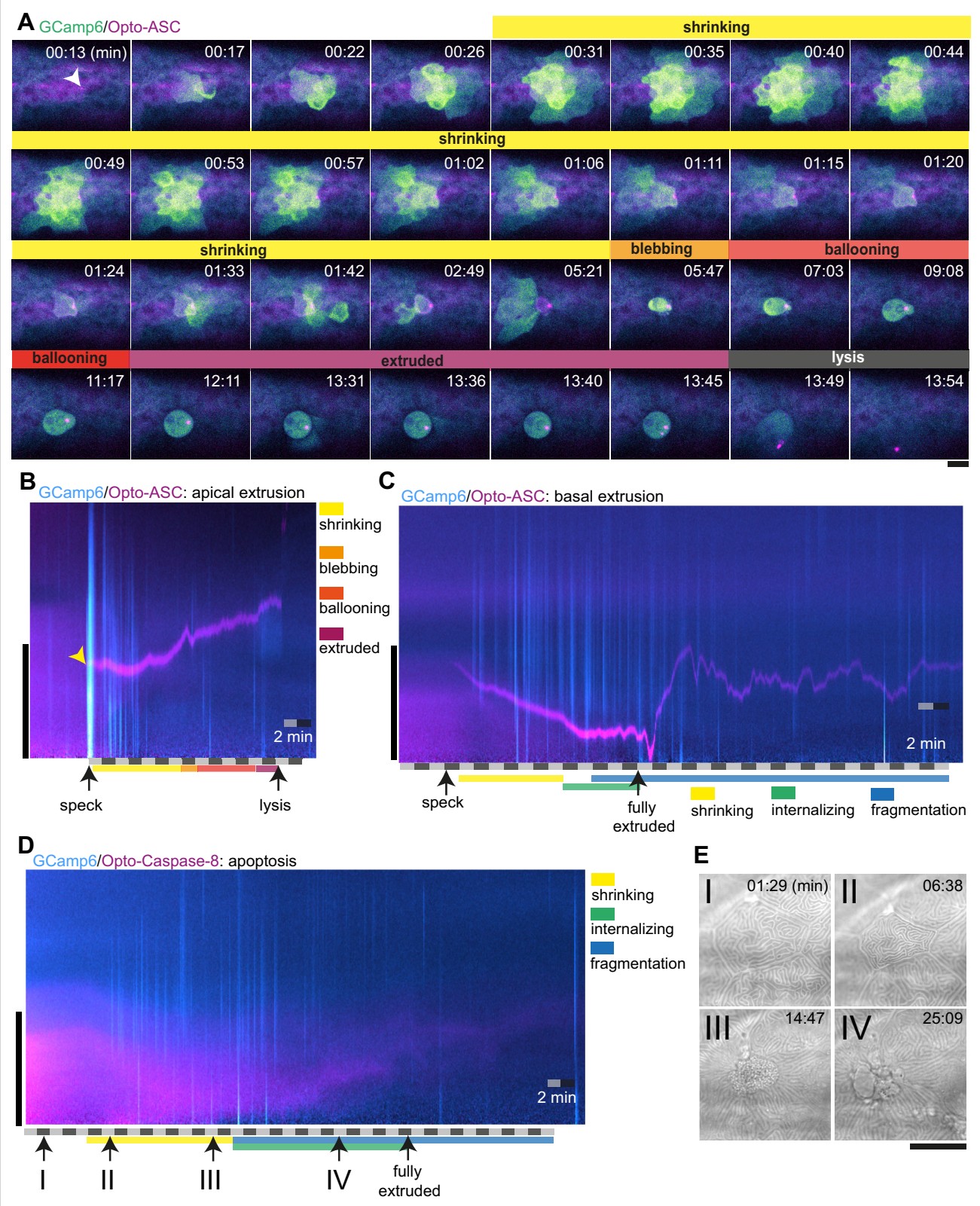

**Figure 8.** Ca²⁺ signaling in epithelial cells in response to dying cells. (**A**) Example of Ca²⁺ signaling in epithelial cells surrounding a cell forming an Opto-ASC speck (yellow arrow head) and being extruded from the tissue. Frames from a time-lapse image sequence of a larva expressing GCamp6 (light green) and Opto-ASC (magenta). Phases of the extrusion were characterized using the bright field channel, as described in *Figure 5* and are color coded (yellow-black). Scale bar is 20 μm. (**B–D**) Two-dimensional representation in time and space (derived from 3D kymographs) of the Ca²⁺ response

*Figure 8 continued on next page*

*Figure 8 continued*

to cell death in the larval epidermis. The y-axis shows the radial space around the dying cell, as described in *Figure 8—figure supplement 1*. GCamp6 Ca$^{2+}$ sensor signal intensity is shown in cyan and Opto-ASC in magenta. The scale bar on the y-axis is 20 µm. Time is marked along the x-axis in 2 min blocks. (**B**) Ca$^{2+}$ signaling response to Opto-ASC-induced apical extrusion as shown in A. The phases of extrusion are color coded (yellow to dark red) as in panel A and *Figure 5C*. (**C**) Opto-ASC-induced cell death with basal extrusion. Phases of extrusion marked along the x-axis. (**D**) Opto-Caspase-8 (magenta)-induced apoptosis. Numbers on the x-axis refers to the images in E. (**E**) Bright field images of the cell analyzed in D illustrating stages of apoptotic cell death. I: Cell morphology before the first signs of cell death; II: isomorphic shrinkage of cell; III: microridge pattern has dissolved and cell starts to internalize; IV: cell has started to fragment. Scale bar in all images is 20 µm.

The online version of this article includes the following figure supplement(s) for figure 8:

**Figure supplement 1.** Method for scoring calcium signalling in time and space and control for background signalling.

**Figure supplement 2.** Ca$^{2+}$ response of neighboring cells to Opto-ASC induced basal extrusion and its quantification.

sporadic Ca$^{2+}$ signaling. We counted the number of responding cells in the ring of cells directly adjacent to the dying cell, and the next ring beyond those. We detected on average 21 basal cells and 12 periderm cells responding to Caspase-8-induced cell death with increased Ca$^{2+}$ levels within the first 15 min after the cell started to shrink (N=3 cells of different larvae; *Figure 8—figure supplement 2C*). When periderm cells died via ASC-induced apoptosis in Caspb knock-outs, the Ca$^{2+}$ response of surrounding cells and the dying cell itself was similar to the response to Opto-caspase-8-induction in the periderm.

In summary, the response of the surrounding cells depended on the direction of extrusion and on the stimulus that triggered cell death. A strong calcium response in the form of a wave in neighboring periderm cells and underlying basal cells occurred only in the case of ASC-induced extrusion and most prominently when cells were apically extruded.

## Discussion

Our results demonstrate that Opto-ASC is an efficient tool to induce inflammasome formation and ASC-dependent cell death in zebrafish. Compared to over-expression of ASC in our earlier studies (*Kuri et al., 2017*), or infection models (*Forn-Cuní et al., 2019*), it allows a more precise spatial and temporal manipulation of inflammasome activation and cell death. Heat-shock induced expression of Opto-ASC is highly variable between cells within individual larvae and between larvae, which has both advantages and disadvantages. Mosaically distinct ASC levels allow the assessment of the role of ASC levels and the response of neighboring cells within the same experimental animal. If a more uniform expression of Opto-ASC were desired, this could be achieved by the expression of Opto-ASC under the control of tissue-specific promoters combined with Cre-Lox or trans-activator-induced expression (*Knopf et al., 2010*; *Mosimann et al., 2011*; *Gerety et al., 2013*).

Opto-ASC oligomerization is efficiently induced within minutes by constant exposure to 488 nm light. After a single light pulse, specks were often not observed immediately, but appeared with a delay of up to 40 min. This is unexpected, because it is unlikely that Cry2-olig would remain active for so long that dimerization could be delayed for more than a few minutes (*Taslimi et al., 2014*), and one would expect that even a single initial

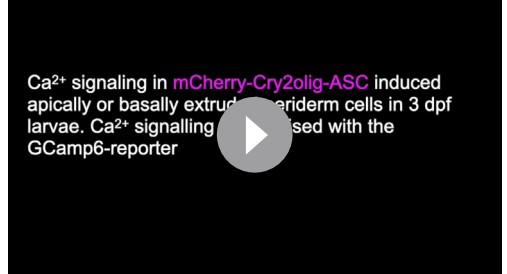

**Video 17.** Ca$^{2+}$ signaling (light green) in epithelial cells surrounding a cell forming an Opto-ASC speck (magenta) and being extruded from the tissue. Time is in minutes.
https://elifesciences.org/articles/86373/figures#video17

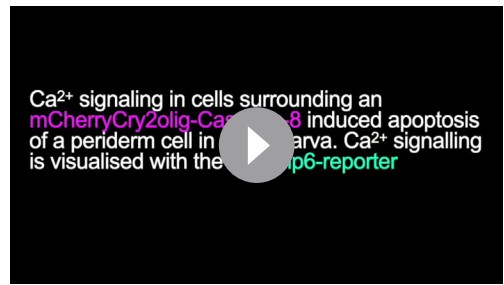

**Video 18.** Ca$^{2+}$ signaling (light green) in epithelial cells surrounding a cell dying apoptotically after Opto-Caspase-8 (magenta) induction. Time is in minutes.
https://elifesciences.org/articles/86373/figures#video18

dimerization of Opto-ASC would have led to the immediate recruitment of the endogenous ASC that is available throughout the cell. Our observations may suggest that the initial seeding event could have created a dimer that was not yet active, but that a second step of maturation could subsequently occur. This highlights the fact that we still know very little about the unusual dynamics of speck assembly, or the mystery why the cell normally always forms only one single speck.

In response to the formation of an Opto-ASC speck periderm cells died in two morphologically different ways. Cells were extruded either apically or basally, and some cells were simultaneously extruded in both ways. Apical extrusion of periderm cells has been described not only in response to infection, but also in response to treatment with the antibiotic geneticin (G418) (*Eisenhoffer and Rosenblatt, 2011*) and during tissue homeostasis as a response to cell crowding (*Eisenhoffer et al., 2012*). Apoptotic stimuli like geneticin have been shown to induce extrusion via sphingosine-1 phosphate (S1P) and its receptor, which is expressed in the surrounding cells. We have previously shown that S1P signaling does not seem to drive extrusion of pyroptotic cells in Caco-2 cell co-cultures (*Shkarina et al., 2022*), but so far it is unclear if S1P signaling or another mechanism activated downstream of the inflammasome is responsible for apical extrusion. Although the factors determining the direction of extrusion remain to be discovered, the different ratios of apical and basal extrusion of periderm cells in different zebrafish strains suggest a genetic component.

These findings indicate that different types of cell death are not entirely separable, but that they represent different manifestations of parallel and interconnected signaling pathways, the overall balance of which defines the final outcomes. This is also evident from other studies in the past, where disrupting one pathway directs the cell towards a different pathway that may have a different phenotypic outcome (*Tsuchiya et al., 2019*). Perhaps it is only through experimental activation of the most downstream effector that a 'pure' phenotype can be induced.

In the case of zebrafish periderm, we identified Caspb as essential for apical extrusion. In the absence of Caspb, cells die with an apoptotic morphology in response to speck formation. The requirement for Caspb for apical extrusion and lysis might also explain the difference in the response to ASC-speck formation between the periderm and the underlying basal cells. According to cell-type-specific RNA sequencing data (*Cokus et al., 2019*), basal cells express ten times lower levels of Caspb than periderm cells.

Although Caspa localizes to the ASC speck (*Kuri et al., 2017*) and is necessary for rapid cell death if no Caspb is present, we did not find it to be necessary for speck induced extrusion. This contradicts our own previous work, in which we identified Caspa as responsible for fast extrusion after speck formation (*Kuri et al., 2017*). We have now re-analyzed our previous data (*Figure 7—figure supplement 2*) and found that the delay between speck formation and cell death can be partially explained by the type of cell that was imaged, namely basal cells, although we observed a much stronger delay between speck formation and cell death in Caspa deficient cells than in wildtype basal cells.

Zebrafish Caspa and Caspb are thus redundant in their role as rapid cell death inducers, but the downstream mechanisms that lead to either extrusion or apoptosis of periderm cells are yet to be identified. These mechanisms may be deployed differently in different situations. For example, in Caco-2 cell monolayers caspase-1 causes pyroptosis with rapid apical extrusion and a lamellipodial response in neighbouring cells which close the wound within approximately 70 min (*Bonfim-Melo et al., 2022*). Wound closure after ASC-induced apical extrusion of zebrafish periderm cells is much faster (around 10 min). Basal cells on the other hand trigger a lamellipodial response in neighbouring cells and take longer to close the gap (*Figure 6—figure supplement 3B*), similar to Caco-2 cells.

Apoptosis and ASC-induced cell death are both initiated within a short time after speck formation and cause the recruitment of actin in surrounding periderm cells to the side facing the dying cells. The closure of the surrounding epithelium above or below the extruded cell is faster for ASC-induced extrusion than for apoptosis. This difference in rate of wound closure could be caused by different dynamics in the cell itself, or in the surrounding cells. The fact that we observe an immediate $Ca^{2+}$ response in the case of ASC-induced death, but not apoptosis, indicates signalling from the dying cell to the surrounding epithelium, and this signal differs between cells dying apoptotically or by ASC-induced death. Caspase-8-induced apical extrusion of MDCK cells is preceded by a strong calcium wave in surrounding cells (*Takeuchi et al., 2020*), further suggesting that the $Ca^{2+}$ wave correlates with the mode of extrusion rather than the trigger for cell death. The fast apical extrusion and the rapid $Ca^{2+}$ response require Caspb, pointing to a Caspb-dependent signalling event. A similar reaction was

observed in cultured intestinal epithelial cells in which a bacterial infection sensed by the NAIP/NLRC4 inflammasome induce a fast myosin-dependent contraction in surrounding cells. This reaction was dependent on the activation of sub-lytic GSDMD pores and the resulting ion flux and independent of the cell death or extrusion of the cell (*Samperio Ventayol et al., 2021*).

Zebrafish do not have a direct homolog of GsdmD but two homologues of GsdmE (Gsdmea/b). Human GsdmE is activated downstream of caspase-3 during apoptosis (*Wang et al., 2017*) and it plays a role in the caspase-3-mediated lytic death of primary human keratinocytes after viral inactivation of the Bcl-2 pathway (*Orzalli et al., 2021*). Both zebrafish gasdermins can be cleaved by Caspb and other apoptotic caspases (but not Caspa) in vitro (*Chen et al., 2021*). It was therefore surprising to find that we did not see a change in peridermal keratinocyte cell death after ASC-speck formation when we knocked out Gsdme a and b by CRISPR/Cas9. also treated larvae with inhibitors of Gasdermin D pore formation but did not see any effect on ASC-speck-induced apical extrusion. Although the CRISPR/Cas 9 strategy is highly efficient, and in our hands has worked well for other genes, we cannot rule out that Gsdme a and b are redundant in their function to mediate Caspb dependent extrusion and that a small amount is sufficient to induce this form of cell death. Additionally, we find that cells which are extruded do not lyse immediately upon swelling which is normally the case in pyroptotic cultured cells (*Shkarina et al., 2022*). Since we find a Caspb-dependent difference in cell death phenotypes, the question arises which other components could cause the apical extrusion of cells and the wave-like $Ca^{2+}$ signaling response in surrounding cells. Possible candidates could be pannexins, especially pannexin1a, which is expressed in skin cells (*Cokus et al., 2019*) and is necessary downstream of caspase-11 to induce pyroptosis in response to LPS in mouse macrophages (*Yang et al., 2015*). The release of ATP caused by pannexin pores might explain the strong calcium wave in the Caspb-dependent extrusion of peridermal keratinocytes (*Mori et al., 2022*).

## Materials and methods
### Zebrafish husbandry, transgenic lines, and genotyping

Zebrafish (*Danio rerio*) were cared for using standard procedures as described previously (*Westerfield, 2000*) and in accordance with EMBL guidelines. All experimental procedures were approved by the EMBL Institutional Animal Care and Use Committee (IACUC nos. 2019-03-19ML). We used the following zebrafish wildtype lab strains: AB2B2, AB, golden and Wild Indian Karyotype (WIK). The *Tg(HSE:mCherry-Cry2olig-asc)* line was generated by co-injecting embryos at the one-cell stage with the *HSE:mCherry-Cry2olig-asc* plasmid with transposase mRNA (100 ng/µl). To follow the actin dynamics in skin cells, we used the *Tg(krt4:Gal4) line* (*Wada et al., 2013*) crossed to *Tg(6xUAS:mNeonGreen-UtrCH_6xUAS:lyn-tagRFP)* generated in the lab of D.Gilmour by Jonas Hartmann. Actin dynamics in basal skin cells were visualized in the *Tg(rcn3:Gal4)* line (*Ellis et al., 2013*) crossed to *Tg(6xUAS:mNeonGreen-UtrCH_6xUAS:lyn-tagRFP)*. To induce specks and follow endogenous ASC we used *Tg(HSE:asc-mKate2)* and the endogenously tagged asc:asc-GFP line (*Kuri et al., 2017*). To induce apoptotic cell death, we used *Tg(HSE:mCherry-Cry2olig-caspase-8*; *Shkarina et al., 2022*). For monitoring $Ca^{2+}$ signaling we used *Tg(ß-actin:GCamp6*; *Chen et al., 2017*).

### Cloning of Opto-ASC variant constructs and transient expression

The Cry2olig-mCherry construct (plasmid #60032) was obtained from Addgene, and plasmids containing *asc*, *asc*(4xmut), *PYD* and *CARD* published in *Kuri et al., 2017* were used as cloning templates. The initial HSE:mCherry-Cy2olig ASC construct was cloned as described in *Shkarina et al., 2022*. Other constructs containing ASC variants (PYD and CARD) used the *HSE:mCherry-Cy2olig-asc* plasmid as background and were modified by seamless cloning using the NEB Assembly Kit, after SmaI/ EcoRV or XmaI/EcoRV digestion of the plasmid (replacing *asc*). The following constructs were generated and used for transient expression: *HSE:mCherry-Cry2oligR489E*, *HSE:mCherry-Cry2olig-PYD* and*HSE:mCherry-Cry2olig-CARD*. All inserts were verified by sequencing, and constructs are deposited in the European Plasmid Repository (https://www.plasmids.eu) For expression of transient constructs, we co-injected expression plasmids with Tol2 transposase mRNA (100 ng/µl) in embryos at the one-cell stage.

Larvae were screened at 2.5 dpf for expression of tag:RFP in the heart as described in *Kuri et al., 2017*; For heat-shock driven expression, we incubated larvae at 2.5–3 dpf in 2 ml tubes in a heating block at 39 °C.

## CRISPR/Cas9-mediated *asc* zebrafish knockout

We designed a custom gBLOCK (Integrated DNA Technologies) incorporating a guide RNA-targeting sequence preceded by a T7 promoter sequence. Guide RNAs against exon 1 of *asc* were used with the following sequences: 5'-GGTGGAGATCGAAGATCAAG-3' and 5'-GCAGCTGCAGGAGGCTTTTG -3' respectively. Guide RNAs were synthesized using MEGA shortscriptTM Kit (Applied Biosystems), according to the manufacturer's protocol, and were purified using RNeasy Mini Kit (Qiagen, #74106). Cas9 mRNA was synthesized using mMESSAGE mMACHINE SP6 Transcription Kit (Thermo Fisher Scientific, #AM1340). 2 nL of a mixture containing 250 ng/µL gRNA and 0.1 mg/mL Cas9 protein was injected into the yolk of one-cell AB zebrafish embryos.

Genotyping was performed by extracting genomic DNA from fin clips or larvae using the QuickExtract DNA extraction solution (Epicentre) and amplifying loci using EMBL in-house Phusion polymerase, primers are listed in the Key Resource Table.

## F0 CRISPR screen

The F0 CRISPR screen was performed as described in *Kroll et al., 2021* for *caspa*, *caspb*, *gsdmea*, *gsdmeb* and *caspase-8a*. sgRNA target sites were selected for high predicted cleavage efficiency within various exons of candidate genes using CCTop (*Stemmer et al., 2015*).

Sequences of synthetic sgRNAs and gene identifiers are listed in the key resource table. We used 3–4 synthetic sgRNAs (Sigma) per gene, which were diluted in water to a concentration of 100 µM. Cas9 protein was produced by the EMBL PEPCore facility and stored in a concentration of 62.5 µM in Cas9 buffer (20 mM Tris-HCl, 500 mM KCl and 20 % vol glycerol). For each experiment, we freshly mixed equal volumes of sgRNA and protein solution (1 µl sgRNA per 1 µl Cas9) and injected the mixture into early embryos within the first 15 min after fertilization. All CRISPR KOs were confirmed by PCR and sequencing around target sites.

## Live imaging and optogenetics

Zebrafish larvae were anaesthetized using 4 mg/ml tricaine pH 7 and mounted on their right side in 0.75% low melting agarose on MaTec glass bottom petri dishes (35 mm No. 1.5 cover glass, 0.16– 0.19 mm). Live imaging was performed at 20 °C using a Zeiss LSM780 NLO confocal microscope (Carl Zeiss) and a 40×C-Apochromat (NA 1.20) water immersion objective (Carl Zeiss). To image GFP, mNeonGreen and the GCamp6 $Ca^{2+}$ reporter, we used the 488 nm multiline Argon laser (Lasos Laser GmbH). To image mCherry and lyn-tagRFP, we used a HeNe laser at 561 nm. For detection of the far-red dye DRAQ7 we used the HeNe 681 laser.

We determined the laser intensity for the 488 nm laser at regular intervals using a chroma slide at a fixed laser power of 5%. The total laser power was measured using a laser power meter (THORLABS GmbH Dachau Germany SN: P0023941) equipped with the sensor S121C 400–1100 nm with a capability of measuring 500 mW. Whenever not stated otherwise, we used 5% 488 argon laser power (24.8microWatt/ $µm^2$) to stimulate oligomerization of optogenetic constructs.

For 2-photon activation of Opto-ASC in single cells, we used a 140-fs pulsed multi photon laser (chameleon; Coherent) of the LSM780 NLO and operated the microscope using the Zen Black Software (Carl Zeiss, 2012 version) together with the Pipeline Constructor Macro (*Politi et al., 2018*).

For overnight imaging of whole larvae, we used a 20 x air objective (Carl Zeiss) and acquired multiple stacks (2x7) to cover the entire larva. Stacks were then stitched together using the Zen Black software. Stacks were acquired every 15 min for 12–16 hr.

For high-resolution time-lapse imaging of the Opto-ASC speck, Opto-PYD and Opto-CARD we used a Zeiss 880 Airyscan and imaged in Airy fast mode with a C-apochromat 40 x/1.2 W Korr FCS M27 objective. For activation of opto-constructs and detection of GFP we used 0.2% 488 nm and for detection of mCherry 1.5% 561 nm. To detect the GCamp6 signal, we optimized imaging conditions to reach a frame rate below 5 sec. The stack size was 8 µm (1 µm z resolution).

## Image analysis and quantification

We used Fiji ImageJ for image analysis (*Schindelin et al., 2012*). Images acquired on the Zeiss 880 Airyscan were processed with Zen black software (Zeiss). We used the cell counter plugin to manually

count total numbers of periderm cells and numbers of periderm cells expressing Opto-ASC. Example images are always presented as maximum intensity z-projections. The total mean fluorescence of larvae was measured in maximum z-projections in the region of interest (ROI) manually defined by the outline of the larva. To determine the expression of Opto-ASC in single cells, we chose the three z-planes in which most of the cell was visible and measured MFI in a selected region with uniform expression in an average intensity z-projection.

Line profiles for high resolution images of specks were normalized by dividing by the average gray value and subtraction of the lowest gray value. For measuring speck size, a line was drawn through the longest diameter of the speck; the diameter was determined by measuring the distance between the two points of maximum slope of the plot profile along the line.

To determine the growth of specks, we marked the outlines of the speck in each time frame for Opto-ASC and ASC-GFP separately and measured the area. We started the measurements at the first time point where mCherry/GFP visibly aggregated. We calculated the time difference starting from the appearance of the first mCherry/mKate2 aggregates to the time point where total intensity (intensity * area) of the speck reached a constant value.

To determine if a periderm cell was extruded apically, basally or in both directions we used bright field imaging. Anything that was retained below the periderm layer was considered as basally extruded, anything that left the tissue and dissociated from the epithelial layer into the surrounding water was considered apically extruded.

For estimating the time between speck formation and initiation of cell death, we calculated the interval from first detectable Opto-ASC aggregates to the time when the cell outline first deformed. For the analysis of $Ca^{2+}$ signaling in bystander cells in response to cell death, we generated kymographs for a circular area around the dying cells using the Radial Reslice Fiji plugin (*Cooper, 2009*) which measures the pixel intensity along a line rotating around a center over 360 degrees. By creating an average projection of all lines, we created a two-dimensional representation of the area around the cell for every time point. The exact procedure with examples is shown in *Figure 8—figure supplement 1*.

To quantify the $Ca^{2+}$ response of bystander cells to basal extrusion and apoptosis, we manually counted the number of periderm cells and basal cells which showed a GCamp6 signal for every minute after initiation of cell death for 15 min.

## Statistics

All statistical analysis was done using Prism version 9 (Graphpad). P-value ranges are included in figures as follows: * is $p<0.05$, ** is $p<0.01$ *** is $p<0.001$ and **** is $p<0.0001$. For comparison of different heat shock conditions (*Figure 2B and C*) and the comparisons in *Figure 7F–H*, we used ordinary one-way ANOVA. For determining correlations, we used simple linear regression. To compare average MFI between speck-forming and non-speck forming cells, we used a Wilkoxon Test. To compare sizes of specks and time of speck formation, we used a Mann-Wittney test. For the difference in time shown in *Figure 4C* and *Figure 5E*, we assumed unequal standard deviations and used a t-test with Welch's correction.

## Chemical treatments

For the detection of cell lysis, we used Draq7 (Invitrogen), a far-red dye that stains nuclear DNA only if membrane integrity is lost. We preincubated fish in 0.3 µM (1:1000) Draq7 in E3 fish medium for 1 hr and added the dye directly to the low-melting agarose and the fish medium in which larvae were imaged, to a concentration of 0.3. µM.

To inhibit GSDMN pore formation, we treated 3 dpf larvae for with 10 µM Disulfiram (Merck) or 25 and 100 µM dimethyl fumarate for 1 hour before and during imaging. Exposure for 2 hr or longer (e.g. over night) was lethal for zebrafish larvae at a dose as low as 1 µM of Disulfiram and 10 µM for dimethyl fumarate (DMF). All inhibitors were dissolved in DMSO.

We also treated 2 dpf larvae over night with 10 µM to 100 µM Gasdermin D inhibitor LDC7559 (MedChemExpress). The inhibitor was added directly to the fish water as described in *Isles et al., 2021*, and larvae were imaged at 3 dpf.

## Acknowledgements

The authors thank the Advanced Light Microscopy Facility (AMLF) at the EMBL-Heidelberg for their continued support and Manuel Gunkel for his help with laser intensity measurements. The authors thank Darren Gilmour and Jonas Hartmann for providing the *Tg(6xUAS:mneonGreen-UtrCH)* zebrafish line. We thank Alexandre Paix for providing guidance and material for CRISPR/Cas9 experiments and Takehito Tomita for help with analysis of GCamp6 imaging data. We thank Girogia Rapti for providing lab space for and Christina Pallares Cartes and Jayan Nair for helping with revision experiments. ML thanks EMBL and the Developmental Biology unit for space and general support, and EMBO for funding. The work in PB's lab was supported by grants from the ERC (ERC2017-CoG-770988-InflamCellDeath), the Swiss National Science Foundation (175576 and 198005), the OPO Stiftung and Novartis.

## Additional information

### Funding

| Funder | Grant reference number | Author |
|---|---|---|
| EMBO | | Maria Leptin |
| European Research Council | ERC2017-CoG-770988-InflamCellDeath | Petr Broz |
| Swiss National Science Foundation | 175576 | Kateryna Shkarina |
| Swiss National Science Foundation | 198005 | Kateryna Shkarina |
| OPO-Stiftung | | Kateryna Shkarina |
| Novartis | | Kateryna Shkarina |

The funders had no role in study design, data collection and interpretation, or the decision to submit the work for publication.

### Author contributions

Eva Hasel de Carvalho, Conceptualization, Data curation, Formal analysis, Supervision, Validation, Investigation, Visualization, Methodology, Writing - original draft, Writing – review and editing; Shivani S Dharmadhikari, Formal analysis, Validation, Investigation; Kateryna Shkarina, Bruno Reversade, Resources, Writing – review and editing; Jingwei Rachel Xiong, Resources; Petr Broz, Resources, Funding acquisition; Maria Leptin, Conceptualization, Supervision, Funding acquisition, Writing - original draft, Writing – review and editing

### Author ORCIDs

Eva Hasel de Carvalho  http://orcid.org/0000-0002-3067-2874
Shivani S Dharmadhikari  http://orcid.org/0000-0003-2804-9079
Kateryna Shkarina  http://orcid.org/0000-0002-2177-4844
Bruno Reversade  http://orcid.org/0000-0002-4070-7997
Petr Broz  http://orcid.org/0000-0002-2334-7790
Maria Leptin  http://orcid.org/0000-0001-7097-348X

### Ethics

All experimental procedures were approved by the EMBL Institutional; Animal Care and Use Committee (IACUC nos. 2019-03-19ML).

### Decision letter and Author response

Decision letter https://doi.org/10.7554/eLife.86373.sa1
Author response https://doi.org/10.7554/eLife.86373.sa2

## Additional files

### Supplementary files
• MDAR checklist

### Data availability
Constructs used to generate transgenic fish lines for this work are deposited in the European Plasmid Repository (https://www.plasmids.eu) under Accession Numbers indicated in the Key Resource Table. Fishlines are available are kept in the EMBL fish facility and are available on request. Sequences of sgRNA used for the F0 KO screen are included in the key resource table. Raw imaging data is available at BioImage Archive (https://www.ebi.ac.uk/bioimage-archive/) under the accession S-BIAD796.

The following dataset was generated:

| Author(s) | Year | Dataset title | Dataset URL | Database and Identifier |
|---|---|---|---|---|
| Hasel de Carvalho E, Leptin M | 2023 | The Opto-inflammasome in zebrafish: a tool to study cell and tissue responses to speck formation and cell death | https://www.ebi.ac.uk/biostudies/BioImages/studies/S-BIAD796 | BioImage Archive, S-BIAD796 |

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

# Appendix 1

## Appendix 1—key resources table

| Reagent type (species) or resource | Designation | Source or reference | Identifiers | Additional information |
|---|---|---|---|---|
| Strain, strain background (Dario rerio) | AB2B2 | Peri & Gilmour lab strain | | EMBL heidelberg Fish facility |
| Strain, strain background (Dario rerio) | AB | ZIRC | | EMBL heidelberg Fish facility |
| Strain, strain background (Dario rerio) | Golden | ZIRC | | EMBL heidelberg Fish facility |
| Strain, strain background (Dario rerio) | WIK Upenn | ZIRC | | EMBL heidelberg Fish facility |
| Genetic reagent (*Danio rerio*) | Opto-ASC: Tg(HSE:mCherry-Cry2olig-asc) | this work | | EMBL heidelberg Fish facility, stable fish line expressing HSE:mCherry-Cry2olig-asc, expression can be induced by heatshock |
| Genetic reagent (*Danio rerio*) | ASCΔ2/Δ2 knockout | this work | | CRISPR/cas9 generated ASC mutant line, deletion of two basepairs creats a frameshift with premature stop codon, Reversade lab fish facility |
| Genetic reagent (*Danio rerio*) | Tg(Krt4:AKT-PH) | *Wada et al., 2013* doi: 10.1016 /j.cub.2013.06.035 | | |
| Genetic reagent (*Danio rerio*) | Tg(6xUAS:mNeonGreen-UtrCH_6xUAS:lyn-tagRFP) | kindly provided by Darren Gilmour and Jonas Hartmann | | |
| Genetic reagent (*Danio rerio*) | Tg(HSE:mCherry-Cry2olig-caspa) | Shkarina et *Shkarina et al., 2022* doi: 10.1083/jcb.202109038 | | available on request, fish are kept at Heidelberg fish facility |
| Genetic reagent (*Danio rerio*) | Tg(HSE:mCherry-Cry2olig-caspase-8) | Shkarina et *Shkarina et al., 2022* doi: 10.1083/jcb.202109038 | | available on request, fish are kept at Heidelberg fish facility |
| Genetic reagent (*Danio rerio*) | Tg(rcn3:Gal4) | *Ellis et al., 2013* doi: 10.1083/jcb.201212095 | | |
| Genetic reagent (*Danio rerio*) | Tg(HSE:asc-mKate2) | *Kuri et al., 2017* doi: 10.1101/111542. | | |
| Genetic reagent (*Danio rerio*) | asc:asc-GFP | *Kuri et al., 2017* doi: 10.1101/111542. | | |
| Genetic reagent (*Danio rerio*) | Tg(ß-actin:GCamp6) | *Chen et al., 2017* doi: 10.1016 /j.ydbio.2017.03.010. | | |
| Recombinant DNA reagent | HSE:mCherry-Cy2olig ASC (Plasmid) | this work | EPR Plasmid #421 | Deposited at European Plasmid Repository (https://www.plasmids.eu) |
| Recombinant DNA reagent | HSE:mCherry-Cy2olig PYD (Plasmid) | this work | EPR Plasmid #422 | Deposited at European Plasmid Repository (https://www.plasmids.eu) |
| Recombinant DNA reagent | HSE:mCherry-Cy2olig CARD (Plasmid) | this work | EPR Plasmid #423 | Deposited at European Plasmid Repository (https://www.plasmids.eu) |
| Recombinant DNA reagent | HSE:mCherry-Cry2oligR489E (Plasmid) | this work | EPR Plasmid #424 | Deposited at European Plasmid Repository (https://www.plasmids.eu) |
| Recombinant DNA reagent | HSE:ASC-mKate2 | *Kuri et al., 2017* doi: 10.1101/111542. | EPR Plasmid #427 | Deposited at European Plasmid Repository (https://www.plasmids.eu) |
| Sequence-based reagent | caspa_F | *Kuri et al., 2017* doi: 10.1101/111542. | PCR Primer (Genotyping) | TGGGTTAACTAGGCAAGTCAGGG |
| Sequence-based reagent | caspa_R | *Kuri et al., 2017* doi: 10.1101/111542. | PCR Primer (Genotyping) | AGGGTGTATCAGGACTTGGGCCC |
| Sequence-based reagent | caspa_R2 | *Kuri et al., 2017* doi: 10.1101/111542. | PCR Primer (Genotyping) | CCACACATGGGAGGTGTGAA |

*Appendix 1 Continued on next page*

*Appendix 1 Continued*

| Reagent type (species) or resource | Designation | Source or reference | Identifiers | Additional information |
|---|---|---|---|---|
| Sequence-based reagent | asc_F | | PCR Primer (Genotyping) | sgRNA sequence:GTCAGCAFCTTCAACGAGAG |
| Sequence-based reagent | asc_R | | PCR Primer (Genotyping) | sgRNA sequence: ACATTGCCCTGTGTTCCTCA |
| Sequence-based reagent | caspa_Ex1_T2 | Merck VC40007 sgRNA | sgRNA | sgRNA sequence: CTGGTACAGGTGGCTCCGGCTGG |
| Sequence-based reagent | caspa_Ex1_T15 | Merck VC40007 sgRNA | sgRNA | TTGGGGCGGATAATCTAAGAAGG |
| Sequence-based reagent | caspa_Ex3_T8 | Merck VC40007 sgRNA | sgRNA | GAACATGGAAAAGCTGTTAAAGG |
| Sequence-based reagent | caspa_Ex4_T3 | Merck VC40007 sgRNA | sgRNA | AATGGCGTCTCTTTTGCCGTGGG |
| Sequence-based reagent | caspb_EX2_T2 | Merck VC40007 sgRNA | sgRNA | AGCAGAACGAACGTGCAAAGCGG |
| Sequence-based reagent | caspb_ EX5_T | Merck VC40007 sgRNA | sgRNA | GGCCTGAATCAGGATAACCTTGG |
| Sequence-based reagent | caspb_EX8_T1 | Merck VC40007 sgRNA | sgRNA | sgRNA sequence: GCAAGAGTTTCGCCTGACCTTGG |
| Sequence-based reagent | gsdmea_Ex1_T1 | Merck VC40007 sgRNA | sgRNA | sgRNA sequence: GTCCGACACATAGATCCTTCAGG |
| Sequence-based reagent | gsdmea_Ex2_T6 | Merck VC40007 sgRNA | sgRNA | sgRNA sequence: AAACTTCTGAACGACACCAGAGG |
| Sequence-based reagent | gsdmea_Ex4_T8 | Merck VC40007 sgRNA | sgRNA | sgRNA sequence: TGTGTTGGCCTACAGTGTAATGG |
| Sequence-based reagent | gsdmea_Ex7_T1 | Merck VC40007 sgRNA | sgRNA | sgRNA sequence: ATGGCGTCTGATCAGAATGGTGG |
| Sequence-based reagent | gsdmeb_Ex1_T12 | Merck VC40007 sgRNA | sgRNA | sgRNA sequence: TCCGGAGCCACCTATATTAATGG |
| Sequence-based reagent | gsdmeb_Ex2_T14 | Merck VC40007 sgRNA | sgRNA | sgRNA sequence: sgRNA sequence:TCCGGAGCCACCTATATTAATGG |
| Sequence-based reagent | gsdmeb_Ex5_T2 | Merck VC40007 sgRNA | sgRNA | sgRNA sequence: TCACAGTTAGTGCTGCACCAGGG |
| Sequence-based reagent | gsdmeb_Ex9_T15 | Merck VC40007 sgRNA | sgRNA | sgRNA sequence: GAAGTGCTGTAGCAGAACTTGGG |
| Peptide, recombinant protein | Cas9 | PepCORE facility, EMBL | | |
| Peptide, recombinant protein | NEBuilder HiFi DNA AssemblyNEBuilder HiFi DNA Assembly Master Mix | NEB | Cat No: E2621 | |
| Chemical compound, drug | Draq7 | Invitrogene | Cat. No.:D15106 | |
| Chemical compound, drug | LDC7559 | MedCehm Express | Cat. No.: HY-111674 | |
| Chemical compound, drug | Disulfiram, European Pharmacopoeia (EP) Reference Standard | Merck | Cat. No.:D2950000 | |
| Chemical compound, drug | dimethyl fumarate | Merck | Cat.No.:242926 | |
| Software, algorithm | Prism version 9 | Graphpad | | |
| Software, algorithm | Fiji distribution of ImageJ | *Schindelin et al., 2012* | | |
| Software, algorithm | Radial Reslice Fiji plugin | *Cooper, 2009* | | |
| Software, algorithm | CCTop | *Stemmer et al., 2015* | | https://cctop.cos.uni-heidelberg.de:8043/index.html |

