## [Editor Report]

This important study describes a novel optogenetic approach to study the responses to inflammasome activation and cell death in zebrafish. The validation of the model is compelling, and its application reveals novel insights in the response to cell death. This new model should help further advance future efforts centered on cell death responses in vivo.

---

## [Decision Letter]

[Editors' note: this paper was reviewed by Review Commons.]

---

## [Author Response]

General Statements [optional]We are obviously very pleased with the general support expressed by the referees, and appreciate their critical comments. We detail below how we propose to respond to their suggestions and queries.In view of the fact that my lab is no longer in existence, I will have to rely on the kind generosity of my colleagues at EMBL to host former team members (the two first authors) for a limited period to come back to Heidelberg to carry out any further experimental work that may be needed. This means we will have to limit the work we can do to those experiments with the highest priority. However, we are optimistic that we will be able to obtain indicative results.We will also follow most of the referees’ other suggestions and requests for additional data and quantifications, as outlined (or already included) below.Description of the planned revisionsReviewer #1 (Evidence, reproducibility and clarity (Required)):Minor comments1. Specific experimental issues that are easily addressable:There's a significant concern with the use of LDC7559 (line 387) as a putative small molecule inhibitor of gasdermin D function to test roles (or lack thereof) of the zebrafish gasdermins in the ASC-triggered lytic cell death responses. A recent study (Amara et al. 2021. Cell. PMID34320407) has reported that LDC7559 does not inhibit gasdermin D (and possibly other gasdermins) but rather acts as an allosteric activator of PFKL (phosphofructosekinase-1 liver type) in neutrophils and thereby suppress generation of the NADPH required for the phagocytic oxidative burst and consequent NETosis. Thus, use of LDC7559 as a presumed gasdermin inhibitor in the current manuscript is problematic and should be deleted. As an alternative pharmacological approach to suppress gasdermin function, the authors might consider the use of either disulfiram (Hu et al. 2020. Nature Immunology. PMID32367036) and/or dimethylfumarate (Humphries et al. Science. 2020. PMID32820063). These would be straightforward additional experiments.

We have ordered the reagents to do these experiments. We are optimistic that we will obtain data that will strengthen this part of the ms and be a pointer for future studies by others.

We propose to keep the information on LDC7559 included, but to discuss the reservations the referee lists above – otherwise, others might ask why we did not even try this inhibitor.

2. Are prior studies referenced appropriately? there are some problems; see below.2a. One paper is cited twice in lines 724-726 and 727-729.2b. Another paper is cited twice in lines 790-792 and 793-795.2c. No journal is included for the referenced study by Shkarina et al. in lines 827-828.2d. No journal is included for the referenced study by Stein et al. in lines 831-832.2e. No journal is included for the referenced study by Masumoto et al. in lines 793-795.2f. No journal is included for the referenced study by Kuri et al. in lines 774-775.

We are embarrassed about these omissions and mistakes and have corrected them..

3. Are the text and figures clear and accurate? Generally, yes but with a few exceptions noted below:3a. line 28: "morphological distinct" should read "morphologically distinct".3b. line 161: this sentence contains in parentheses "for how long?" I think this was a comment by one author that wasn't removed from the final submitted manuscript.3c. line 945: spelling "balck" > "black".3d. line 268: "whereas showed a delayed speck formation": the authors need to specify what model/ condition showed a delayed speck formation.3e. line 262: spelling "egnerated" > "generated".

Thank you, all corrected.

Reviewer #2 (Evidence, reproducibility and clarity (Required)):1. In Figure 3 and Figure 4, the majority of Opto-Asc localizes to the plasma membrane but not endogenous Asc. It seems that tagging affects its localization, which could potentially explain its slow kinetics in oligomerization.

That is an interesting suggestion. The membrane enrichment is indeed reproducible and we have no full explanation for it. However, ASC itself seems to have some affinity for the cell cortex as seen by its association with the apical actin ridges in keratinocytes in the resting state (see e.g. figure 3A). Affinity of ASC for actin is also documented in the literature:(F-actin dampens NLRP3 inflammasome activity via flightless^-1^ and LRRFIP2 OPEN; https://doi.org/10.1038/srep29834).

Perhaps the fusion to the optogenetic module somehow enhances the affinity through the initial dimerization. But we can only speculate and have no further evidence that would allow reliable conclusions.

2. In Figure 7, the authors showed that deletion of Caspb, but not Caspa, affected the apical extrusion, without affecting cell death. This may indicate that other caspases, like Caspase-8 or/and caspase-3 were involved. This could be addressed through deletion of Caspase-8 or/and caspase-3.

These are experiments we had in fact done. Unfortunately, they did not allow us to address the question, because the deletions resulted in embryonic lethality. We have added this information to the text.

3. It is very surprising that Opto-Asc-mediated cell death is not dependent on Gasdermins, at least in Caspb-dependent apically extruded dead cells.

Indeed – but see comment by and our response to reviewer 1. We hope to be able to provide additional data.

Reviewer #3 (Evidence, reproducibility and clarity (Required)):Main points:1. So far, it remains a bit unclear how the authors define precisely speck versus any aggregate and the light induced clusters of Cry2 olig. Is it related to the timescale of formation and/or the lifetime of the aggregates? Is it related to their size?

There Is no ‘formal’ definition of an inflammatory speck apart from it being the unusually large aggregates that ASC forms once it is activated. Light-induced clusters of Cry2Olig alone, or of Cry2olig fusions with proteins that do not normally oligomerize are much smaller (extensive documentation in the literature).

A speck is thus a stable aggregate of ASC which is usually around 1 µm in size and is able to activate downstream caspases. But neither of these aspects alone are unique to ASC: prion-like structures can also be large aggregates (indeed ASC-specks have been compared to prions), and much smaller molecular assemblies can activate caspases. Thus ‘speck’ is more an operational definition, and ‘natural’ specks do have both of these properties, but as our experiments show, the properties can actually be separated. I would rather not try to narrow or change the definition, but leave any further discussion to the experts in the field.

Figure 4E shows a number of variants of ‘speck’-like and other multimers: ASC-mKate and Opto-ASC form large single specks in the presence of endogenous ASC. Opto-ASC specks are only slightly smaller than those formed by endogenously tagged ASC-GFP (see also Supplementary Figure 2E). Opto-PYD recruits endogenous ASC and becomes incorporated into a speck of approximately the same size, while Opto-CARD does so less efficiently. All of these kill cells. In the absence of endogenous ASC, Opto-ASC forms much smaller specks, and very many in each cell, but these are still functional as seen by the fact that they still kill cells (not the large spot at t = 60 min in the right half of Figure 4E is not a speck, but the contracted dying cell). Both Opto-PYD and Opto-CARD also form only the small aggregates (quantification will be included), with Opto-PYD still killing the cell by virtue of its ability to recruit caspases via their PYD, whereas Opto-CARD does not.

Since the authors use most of the time constant blue light illumination, could they also assess how long the speck remains after stopping blue light exposure and quantify their lifetime (relative to the CRY2olig cluster lifetime)?

Briefly, any speck that contains a functional ASC moiety remains stable and does not disassemble once the blue light is turned off. In skin cells it is not possible to make quantitative measurements because they are killed by the speck. Opto-ASC specks remain stable until they are taken up by macrophages, as originally reported for ASC-GFP specks in Kuri et al. 2017.

Stability can best be assessed in muscle cells, which do not die upon speck formation. The figure below shows that specks begin to form within minutes of a short pulse of illumination and remain stable (and indeed grow further) for at least 60 min.

Author response image 1 has an example:

**Author response image 1. sa2fig1:** Stability of Opto-ASC specks in muscle cells after exposure to a single pulse of blue light specks in muscle cells expressing Opto-AscTg(mCherry-Cry2olig-asc) are induced by a single illumination with blue light (488nm) at t = 0 for 32 seconds. Multiple oligomers begin to form within 6 minutes, continue to gradually increase in number and, and remain until the end of the movie (60 mins). Cell outlines in the overlying epithelium labeled by AKT-PH-GFP are faintly visible in the first frame. Scale bar is 20 mm.

Similarly could they provide some comparison of the size and localisation of CRY2 olig clusters compared to the speck.

For size, see above. In addition, the size of the Cry2 oligomers as well as of Opto-ASC specks can vary with expression levels.

For location, Cry2olig clusters are usually distributed throughout the cell, as seen in most of the right panels in Figure 4E, and in earlier work in cultured cells (e.g. Taslimi et al. 2014). ASC specks can form anywhere in the cell, while Cry2olig-ASC has a preference for the cell cortex, but this is not absolute. In keratinocytes, but not in basal cells, the speck usually forms close to the lateral membrane. In the absence of endogenous ASC no real speck is formed but Opto-ASC in this case shows no clear localisation of Opto-ASC to the membrane.

In view of the variation we see, a strict quantification is difficult: what would be the ‘correct’ definition of classes to look at? To make statistically significant statements, we would need an enormous number of examples in which we could control for all of the variation of expression levels, cell size, day to day variation etc, and we currently don’t have these. We hope the qualitative evidence in the micrographs we show represents the differences well, and we will be happy to provide a larger number of images, if the referees feel this would be helpful.

With the non functional CRY2olig Asc fusion (Cter fusion), do they still see transient olig2 clustering which then reverse when blue light illumination is gone? I think it might be useful to clarify these points in the main text since most of the quantifications are based on speck localisation/numbering, so their characteristics have to be very well defined.

That would be interesting to work out, but after our initial experiments with this construct, we did not pursue this further, since it was not a pressing issue at the time. If we can fit this into our planned experimental time table, we will re-assess it. However, while of interest, we feel these data would not add substantially to what we know at this point.

2. In all the snapshots of speck formation, there seems to be a relative enrichment of the ASC signal at the cytoplasmic membrane (relative to the cytoplasm) prior to strong speck formation. This seems specific of optoASC as it does not seem to happen for the endogeneous ASC or upon overexpression of ASC-mKate (both in this study and in the previous study published by the same group). Is this apparent membrane enrichment something reproducible? (I see that on pretty much every example of this manuscript). If so what could be the explanation? Is there an actual recruitment at the membrane or is it because the membrane/cortical pool takes longer to be recruited in the speck (hence looking relatively more enriched at intermediate time points).

See our speculations in response to point 1 of the first referee.

We too would really like to understand this, but see no easy and efficient way of testing it at this point.

3. There is also a very distinctive ring accumulation that seems to match with apical constriction and/or a putative actomyosin ring (since this is perfectly round, it could match with a structure with high line tension) (see Figure 1E, Figure 3B, Figure 4D…). Is it something already known? Could the authors comment a bit more on this? This could suggest that ASC accumulates in actomyosin cortex, which would be a very interesting property.

We see that we had failed to be clear about this.

There are two types of actin-labelled rings that appear around dying cells. One is formed by the epithelial cells that surround the dying cell. This structure becomes visible as soon as the cell begins to shrink. That it is formed by the surrounding cells is clear from mosaics where the dying cell does not express the actin marker (e.g. suppl. Figure 4A) and the parts of the ring are seen only in the subset of surrounding cells that do express the marker. This ring is also not circular, but follows the polygonal shape of the shrinking cell. We believe that this is the contractile structure that closes the wound, as observed in many other cases of wound healing.

The other is the one the referee describes here. It is formed within the dying cell, as shown by the fact that it is visible in labelled cells when all the surrounding cells are negative for the marker. The other difference is that it appears only once the dying cell has already contracted considerably and begins to round up and be extruded (most clearly seen in Figure 1E). The third referee had raised a similar point in relation to the same structure seen in Figure 6C, and we provide below the requested analysis. It relies on resolution in the y-axis, which is unsatisfactory, but nevertheless, it is clear that this ring is in a plane above the apical surface of the epithelium (marked by the red membrane marker, i.e is present in the detaching cell). It may well simply be actin appearing in the entire cortex of the cell as it rounds up and looking like a ring when seen from above. A completely different method for imaging would have to be set up to document this reliably, but we hope that these explanations help to clarify the confusion we may have created.

We do not see this accumulation in cells that leave the epithelium towards the interior (see figure in the response to ‘minor points’ below).

4. In the end, since cell death can also occur without visible speck formation, I am wondering if they are eventually the most relevant structure to be quantified. Is it because speck can be dissolved upon caspase activation and could it relates to the speed at which caspase are activated (which may not leave enough time for strong aggregation and visible speck formation)? I believe it would help to get more explanation/discussion on this point.

As already mentioned above, it is indeed not obvious what the significance of the large speck is (and it is extremely puzzling why it is that normally one a single one forms in each cell). We agree that it is not necessarily functionally relevant for the signalling outcome to quantify this property – but nor was this the purpose of this work. Regardless of what kind of aggregate is formed, the optogenetic tool allows the induction of ASC-dependent cell death, and therefore the study of the ensuing cellular events.

5. The compensatory mechanisms that lead to cell death/extrusion despite depletion of caspb is very interesting. Could the authors use some pan caspase inhibitor (like zvad FMK) to confirm that this block opto-ASC cell death also in this context? Alternatively could they check the status of effector caspase activation using live probe (nucview) or immunostaining in the context of caspb depletion?

Those would be interesting avenues to pursue. However, for the reason stated above (Leptin lab closing down, members of fish group no longer at EMBL), we are forced to restrict ourselves to the most important experiments, and think we should prioritize the ones mentioned above.

6. If I understand well, Figure 7C on the right side suggest that the double KO cells don't extrude (if indeed "no change" mean no extrusion, by the way this nomenclature may deserve some clarification in the legend). I don't think these results are mentioned at any point in the main text, but it would be important to include them (since this is an important control).

This interpretation is in fact correct, and we have changed the labelling in the figure to ‘no immediate death’

7. Waves of calcium following cell death and cell extrusion have been previously characterised (Takeushi et al. Curr Biol 2020, Y Fujita group). Interestingly, in this previous article they observed waves of calcium near Caspase8 induced death (in MDCK) as well as near laser induced death in zebrafish, while apparently the authors don't see such Calcium waves upon Caspase8 activation in the zebrafish here. I think it would be important to include a comparison of the authors results with this previous paper in the discussion

We have included this in our discussion.

8. There is also a previous study which characterised the impact of caspase1 on cell extrusion (Bonfim Melo et al. Cell Report 2022, A. Yap lab) which promotes apical extrusion in Caco2 cells. I think it would also be important to include this work in the discussion and to compare with the results obtain here in vivo.

We have included this in our discussion.

Other minor points:1. Line 439: are the numbers given in percentage? if these are absolute numbers, it is out of how many cells ? Same remark line 445: what are the number of cases representing? (percentage?)

We have rephrased this to make it unambiguous.

2. Figure 5: could the authors show periderm and basal cell extrusion with the same type of markers? (membrane or actin or ZO1)? This would help to really compare accurately the morphology and the remodellings associated.

We used Utr-mNeonGreen to lable actin both in periderm and basal cells. Actin labeling of extruded periderm cells is shown in figure 6C, actin labeling of a dying basal cells and the overlying periderm cells is shown in supplementary figure 5A.

3. Is there any obvious differences in cell size or characteristic cell shape between the classic lab strains (golden, AB, AB2B2) and the WIK and experiment strain used here? I do acknowledge that this is clearly not the focus of this study, but given this striking difference (which is related to an important question in the field of extrusion), it would interesting to mention this if there is anything obvious.

We will make these measurements and include the data.

4. Figure 6C: what is exactly the localisation in Z of this strong actin accumulation observed during apical extrusion? Is it apical or is it rather on the basal side of the cell? A lateral view of actin could be useful in this figure for all the different conditions described.

See response to ‘main point 3’ above.

The images that show this are below. However, even from these images it is hard to appreciate the locations. They are in fact much easier to see by following the movies over time, and through the z-sections at any given time point. We will of course submit the movies with the manuscript.

**Author response image 2. sa2fig2:** Localization of actin in the yz and xz planes in Opto-Asc-induced cell death and Opto-caspase-8-induced apoptosisOrthogonal projections of images of apically (A) and basally (B, C) extruded cells at four time points from time lapse recordings. Each time point shows the x-z plane and the orthogonal yz and xz planes, in which the apical sides of the epithelium faces the x-z image. Actin is labeled with mNeonGreen-UtrCH (cyan), plasma membranes and internal membranes by lyn-tagRFP (magenta). Actin is initially concentrated in the apical cortical ridges of periderm cells.A. Apically extruded cell after death is induced by Opto-Asc. As the cell dies actin is lost from the apical ridges and accumulates in the cell cortex in a plane above the original apical surface of the epithelium.B. Basally extruded cell after death is induced by Opto-Asc. Actin is retained in the apical ridges as the cell shrinks and moves below the epithelium within the dying cell.C. Basally extruded cell after death is induced by Opto-Caspase 8. The apical surfaces forms a transient dome in which the actin ridges remain intact before the dying cell is internalized.

5. Figure S3B: could the authors show the utrophin-neonGreen channel separatly? Is there a ring of actin in the dying cell? Also are the membrane protrusion formed more basally? (I suspect this is a z projection, but this would need to be specified in the legend).6. Figure 4A legend: I guess the authors meant red arrowheads rather than frame ?

This has been corrected

7. I list below a number of typos I could find in the main text

Thanks for noticing these, we have corrected all of these, as well as further typos we found.

Line 29: in.Line 30: but.Line 151 : from the …[…] (tissue ?)Line 161: there is most likely a text commenting that was not removed (for how long?)Line 262: generated (egnrtd).Line 268: whereas showed a delay (the subject is missing).Line 269: a point is missing.Line 362: which the lack.Line 368: a point is missing.Line 400: a space is lacking "cellsdepending".Line 438: shrinkwe (space).Line 459 : or I infections.Line 525: there is a point missing.Description of the revisions that have already been incorporated in the transferred manuscriptWe have thoroughly re-edited the text, not only for the typos etc. mentioned by the referees, but also for other aspects of grammar, ambiguity etc. We also added text on the additional points in the main text and discussion as suggested by the referees. We will obviously have to do a final edit once we have carried out the additional experiments.Description of analyses that authors prefer not to carry outAs indicated in the introductory note, we would love to continue this project, including in the directions the referees propose, but we will have to limit ourselves to those that are most important to strengthen the points we make. The experiments mentioned by referee 3 under point 5 would take some testing of conditions and reagents, and while we will give it a try while the two first authors will have the opportunity to return to EMBL for additional experimental work, we can only do this if the experiments with higher priority will work as smoothly and efficiently as we are hoping they will. And we agree with the referee that this is an interesting avenue to pursue, but feel it is not essential to support the validation of the tools we describe or support the other results we present.5. The compensatory mechanisms that lead to cell death/extrusion despite depletion of caspb is very interesting. Could the authors use some pan caspase inhibitor (like zvad FMK) to confirm that this block opto-ASC cell death also in this context? Alternatively could they check the status of effector caspase activation using live probe (nucview) or immunostaining in the context of caspb depletion?

Those would be interesting avenues to pursue. However, for the reason stated above (Leptin lab closing down, members of fish group no longer at EMBL), we are forced to restrict ourselves to the most important experiments, and think we should prioritize the ones mentioned above.